# Inflammatory conditions shape phenotypic and functional characteristics of lung-resident memory T cells in mice

Anna Schmidt[1], Jana Fuchs[1], Mark Dedden[2], Katharina Kocher [3], Christine Schülein[3], Julian Hübner [1,4], Ana Vieira Antão[1], Pascal Irrgang[1], Friederike Oltmanns[1], Vera Viherlehto[1], Natascha Leicht[5], Ralf Joachim Rieker[5,6], Carol Geppert[5,6], Uwe Appelt[7], Sebastian Zundler [2], Kilian Schober [3,8], Dennis Lapuente [1] & Matthias Tenbusch [1,8] ✉

Lung tissue-resident memory T cells ($T_{RM}$) are critical for the local control of respiratory tract infections caused by influenza A viruses (IAV). Here we compare $T_{RM}$ populations induced by intranasal adenoviral vector vaccines encoding hemagglutinin and nucleoprotein (NP) with those induced by an H1N1 infection in BALB/c mice. While vaccine-induced $T_{RM}$ express high levels of CD103 and persist longer in the lung parenchyma, short-lived, H1N1-induced $T_{RM}$ have a transcriptome associated with higher cytotoxic potential and distinct transcriptional profile as shown by single-cell RNA sequencing. In both the vaccine and H1N1 groups, NP-specific CD8$^+$ T cells expand during heterologous influenza virus infection and protect the mice from disease. Meanwhile, lung inflammation in response to an infection with unrelated respiratory syncytial virus do not influence the fate of pre-existing $T_{RM}$. Our preclinical work thus confirms that inflammatory conditions in the tissue shape the phenotypic and functional characteristics of $T_{RM}$ to serve relevant informations for optimizing mucosal vaccines.

Influenza viruses are a common cause for severe respiratory tract infections resulting globally in about 290,000-650,000 deaths per year[1]. Currently, available trivalent or quadrivalent influenza vaccines (TIV/QIV) induce short-lived, strain-specific antibody responses against the glycoproteins hemagglutinin (HA) and neuraminidase (NA). These provide efficient protection against infection with influenza A virus (IAV) strains homologous to the vaccine strains but show limited efficacy against strains evolving from constant mutational changes (antigenic drift)[2,3]. The lack of heterosubtypic protection might also be attributed to the limited capacity of current vaccine formulations to generate virus-specific T-cell responses. Pre-existing T-cell immunity to more conserved antigens, such as the nucleoprotein (NP) or the viral polymerase complex, has been shown to confer protection against severe disease in humans and mice[4–6]. In particular, the induction of tissue-resident memory T cells ($T_{RM}$) upon infection at the viral entry site, the respiratory mucosa, contributes to rapid control of

---

[1]Friedrich-Alexander-Universität Erlangen-Nürnberg, University Hospital Erlangen, Institute of Clinical and Molecular Virology, Erlangen, Germany. [2]Department of Medicine 1, University Hospital Erlangen and Friedrich-Alexander-Universität Erlangen-Nürnberg, Erlangen, Germany. [3]Mikrobiologisches Institut – Klinische Mikrobiologie, Immunologie und Hygiene, Universitätsklinikum Erlangen und Friedrich-Alexander-Universität Erlangen-Nürnberg, Erlangen, Germany. [4]Medizinische Klinik und Poliklinik II, Lehrstuhl für Zelluläre Immuntherapie, Universitätsklinikum Würzburg, Würzburg, Germany. [5]Institute of Pathology, University Hospital Erlangen and Friedrich-Alexander-Universität Erlangen-Nürnberg, Erlangen, Germany. [6]Comprehensive Cancer Center Erlangen-EMN (CCC), University Hospital Erlangen, Friedrich-Alexander-University Erlangen-Nürnberg, Erlangen, Germany. [7]IZKF, Nikolaus-Fiebiger-Centre of Molecular Medicine, Friedrich-Alexander-Universität Erlangen-Nürnberg, Erlangen, Germany. [8]FAU Profile Center Immunomedicine, Friedrich-Alexander-Universität Erlangen-Nürnberg, Erlangen, Germany. ✉e-mail: matthias.tenbusch@fau.de

viral replication in case of secondary infections with heterosubtypic strains[7–10]. Repair-associated memory depots (RAMDs) are seen as a primary niche for CD8$^+$ T$_{RM}$, whereas CD4$^+$ T$_{RM}$ are predominantly found in inducible bronchus-associated lymphoid tissues (iBALTs)[11,12]. iBALT formation is initiated by various inflammatory events or viral infections close to the basal side of the bronchial epithelium. iBALTs represent a type of tertiary lymphoid structure consisting of B-cell and CD4$^+$ T-cell clusters that facilitate direct B-cell help but may also contribute to local T-cell maintenance[13–15]. In the lung, CD8$^+$ T$_{RM}$ and partly CD4$^+$ T$_{RM}$ have been identified by the expression of CD69 and/or CD103[16,17]. The reactivity of this resident CD8$^+$ T-cell population is reflected by their high expression levels of cytotoxic molecules as well as through the presence of preformed mRNA coding for pro-inflammatory cytokines like interferon-gamma (IFNγ)[18,19]. Thus CD8$^+$ T$_{RM}$ can mediate direct lysis of infected cells but also induce a local antiviral state by secretion of pro-inflammatory cytokines and chemokines. This activates the surrounding epithelial tissue and results in the recruitment of other immune cells, such as natural killer cells (NK cells), monocytes, neutrophils, and circulating memory T cells[20,21]. Beyond this cell-mediated effect, it has previously been shown that the activation of CD8$^+$ T$_{RM}$ triggers vascular permeability and, consequently, enables the rapid distribution of serum antibodies into the local tissue, pointing out a possible synergy between CD8$^+$ T cells and humoral immunity[22].

Based on the knowledge derived from infection-induced T$_{RM}$, several groups successfully reported on vaccine-induced T$_{RM}$ providing cross-protection against heterosubtypic IAV strains in animal models[23–26]. For example, lung-resident memory T cells were efficiently induced by intranasal (i.n.) immunization with recombinant viral vectors, such as adenoviral vectors (rAd) or modified vaccina Ankara (MVA) encoding for NP or matrix 1 protein[23,24,27]. We and others demonstrated that local antigen expression and inflammation are prerequisites for the recruitment of antigen-specific T cells and the imprinting of the T$_{RM}$ phenotype[11,12,23,28]. Furthermore, RNA-sequencing experiments revealed that T$_{RM}$ signatures vary depending on the inductive tissue and that T$_{RM}$ exhibits their own organ-specific transcriptional networks to establish tissue residency[29]. However, it is not fully understood how the initial inflammatory conditions shape the phenotypic and functional characteristics of the induced T$_{RM}$ population within the same tissue.

Here, to address this issue, we compare lung T$_{RM}$ populations induced by either a mucosally applied adenoviral vector vaccine or by primary IAV infection. The two treatments provoke different patterns of cytokine production and cell migration in the lung tissue shaping different inflammatory environments during the priming of antigen-specific T$_{RM}$. The newly primed T$_{RM}$ populations display differences in regard to their longevity, localization within the tissue, and their functional and transcriptional profiles. Despite the heterogeneity in the T$_{RM}$ compartment, both mucosal vaccinations as well as primary infection provide heterosubtypic protection against an H3N2 infection even seven months after treatment. Thus, we demonstrate the importance of T$_{RM}$ in protective mucosal immunity in the mouse model but also emphasize the importance of a better understanding of T$_{RM}$ biology for future vaccine development against respiratory viruses.

## Results
### Longitudinal phenotypic and functional analyses of lung-resident memory T cells
The differentiation of lung T$_{RM}$ is driven by the local inflammatory milieu and local antigen presentation. To compare the phenotype and functional profile of T$_{RM}$ triggered either by adenoviral vector immunization or IAV infection, BALB/c mice were intranasally immunized with rAd encoding full-length HA and NP adjuvanted with rAd-IL-1β (rAd-HA/NP/IL-1β) or infected with H1N1 A/PR/8/34 (H1N1),

respectively. Control mice remained untreated (naïve). As part of our hypothesis, the local inflammatory responses were characterized by measuring the levels of antiviral cytokines/chemokines in lung homogenates and the presence of infiltrating immune cells in the BALF during the first two weeks after both treatments (Supplementary Figs. 1, 2). There are significant differences in the cytokine profiles and the kinetics of the innate as well as the adaptive immune cell migration. In the adenoviral vector-treated animals, high levels of IL-1β were detectable at day one, which then declined until day 3, potentially reflecting the vector-driven expression, before a secondary wave of potential endogenous IL-1β were seen on day 7. However, this early inflammatory response also included the production of high levels of CXCL1, GM-CSF, IL-6, CXCL10, CCL2, and IFNγ, which were not present in the H1N1-infected animals at that early time point but partially appeared during the ongoing viral replication. In stark contrast, type I Interferon (IFNα/β) production was significantly higher in the infected animals and almost absent in the rAd-immunized mice except some IFNβ at day 7. Overall, the peak of inflammation seemed to be reached in both groups at day 7 post-treatment with declining levels of most cytokines thereafter (Supplementary Fig. 1). In line with high levels of CXCL1, huge amounts of neutrophils were present in the BALF beginning from day one after rAd treatment (Supplementary Fig. 2). In contrast, only few neutrophils were detectable in H1N1-infected animals at day 7, at which also the maximum numbers of monocytes, NK, B and CD8$^+$ T cells were measured. Interestingly, the number of B cells and CD8$^+$ T cells already declined from day 7 to day 15 in the H1N1 group, whereas there was still a continuous increase in this lymphocyte populations in the rAd group (Supplementary Fig. 2). Here, significant differences in the local inflammatory milieu were confirmed after H1N1 infection and adenoviral vector immunization.

To obtain insights into the phenotypic states of CD8$^+$ cytotoxic T lymphocytes (CTLs) and CD4$^+$ T helper cells (Th) triggered by vaccination or infection, lymphocytes were isolated from lung tissue and analyzed by flow cytometry at various time points (Fig. 1 A). HA$_{533-542}$- and NP$_{147-155}$-specific CD8$^+$ T-cell responses were identified by peptide-MHC pentamer (Pent)-staining and further characterized by the expression of tissue residency markers, such as CD69 and CD103. To discriminate blood-derived and tissue-resident memory T cells more unequivocally, intravascular staining with labeled anti-CD45 antibodies was performed (Fig. 1B, C, gating in Suppl. Fig. 4). Two weeks after the treatment, in both experimental groups, substantial amounts of HA- and NP-specific CD8$^+$ T cells were found in the lung, which were predominantly linked to the T$_{RM}$ compartment (iv$^-$; Fig. 1B). In the following contraction phase, the number of HA-specific cells declined with similar kinetics in both groups and only few memory cells were found at the later time points (100 dpi / 150 dpi). However, NP-specific CD8$^+$ T cells induced by adenoviral vector immunization persisted for a longer period of time and were significantly greater in numbers compared to the ones found in previously infected animals throughout the memory phase (Fig. 1B). A closer look at the different memory sub-populations, revealed substantial differences between the two treatment groups (Fig. 1C). Although the most abundant Pent$^+$ memory phenotype was the CD69$^+$CD103$^+$ T$_{RM}$ in both groups at an early time point, there was a more rapid decline of the T$_{RM}$ compartment in the previously infected animals, which occurred for both antigen specificities (Fig. 1C). There were hardly any T$_{RM}$ detectable 150 days after the primary H1N1 infection. Contrary, about 90% of all NP-specific CD8$^+$ T cells found in the lungs of adenoviral vector-immunized mice were tissue-resident at this late time point. Interestingly, these memory cells showed dominant expression of the T$_{RM}$ marker CD103 but lacked the expression of CD69.

To test whether the different phenotypic appearance of T$_{RM}$ coincided with differences in the functional T-cell profile depending on the priming event, we next investigated cytokine production after antigenic stimulation. Interestingly, the frequency of CD8$^+$ T cells

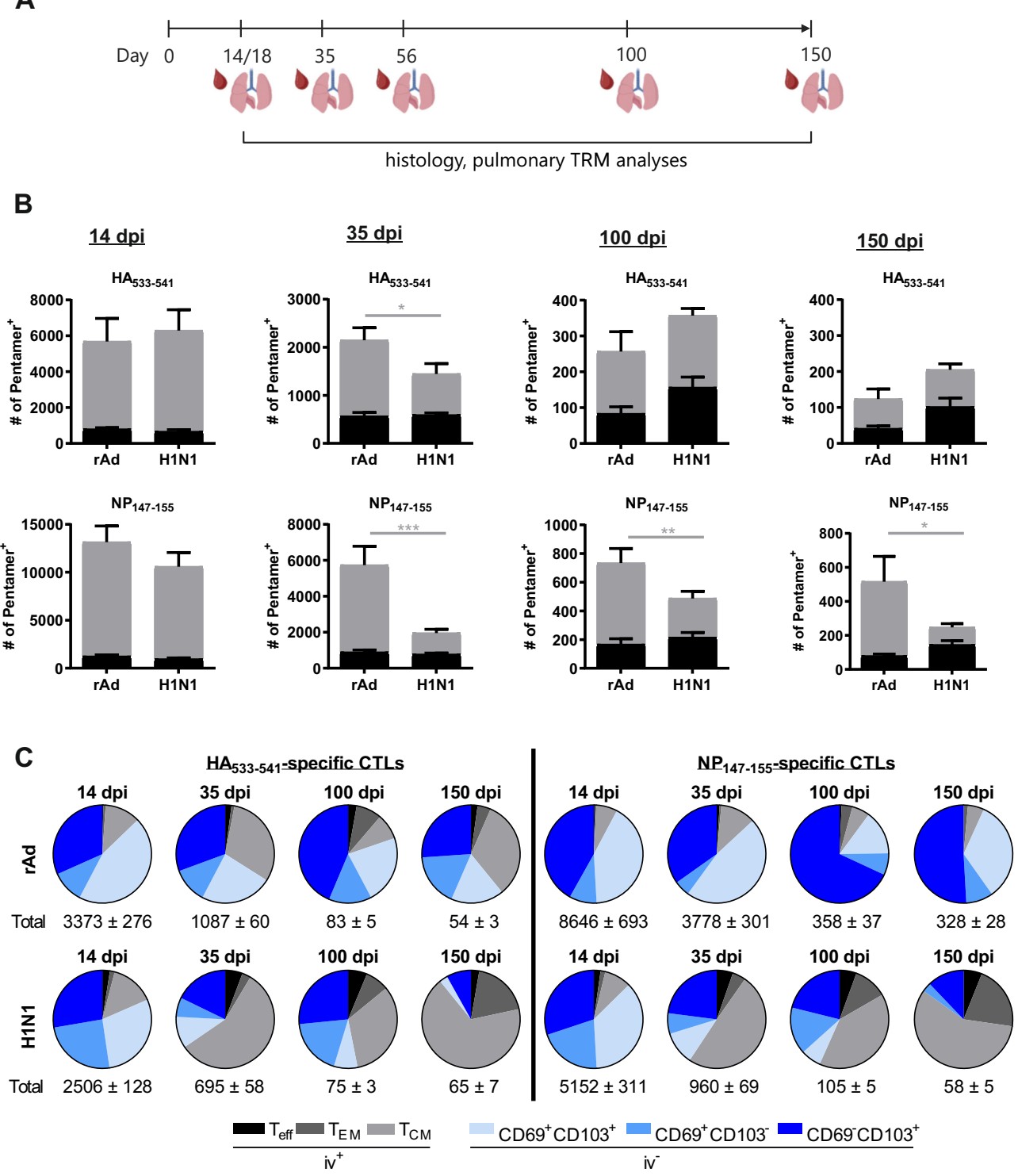

**Fig. 1 | Schedule of experimental treatments and kinetic phenotypic analysis of antigen-specific CD8+ T_RM. A** On day 0, seven-weeks-old female BALB/c mice were either i.n. immunized with rAd-HA, rAd-NP, and rAd-IL-1β (each 2 × 10^8 particles) or infected with the H1N1 strain A/PR/8/34 (100 PFU). At the indicated time points, lymphocytes were isolated from lung tissues to evaluate T-cell phenotypes and functional responses. In addition, blood and BALF samples were collected at the specified times to assess systemic and local antibody responses. The localization and organization of different immune cells was examined by histological techniques. **B** At the indicated times, antigen-specific cells were identified by pentamer staining, and total numbers of HA_533-541- and NP_147-155-specific Pent+ cells are depicted. The distribution of antigen-specific T cells within the iv- and iv+ proportions are illustrated by different coloring. **C** Absolute numbers of circulating effector and memory T-cell subsets, and those of the T_RM pool induced by mucosal vaccination or H1N1 infection are shown. Iv- T_RM (KLRG1-) are colored in different shades of blue (regarding CD69 and CD103 expression), while effector T cells (T_eff; iv+ KLRG1+ CD127-) are shown in black, and effector memory T cells (T_EM; iv+ KLRG1 + CD127 +), and central memory T cells (T_CM; iv+ KLRG1- CD127 +) are shown in gray. Each data set represents the mean + SEM of per group (n = 6 mice per group/timepoint, except 150 dpi in the H1N1 group represents only n = 4). Statistical significances were analyzed by two-way ANOVA followed by Šídák's multiple comparison test (*, p < 0.1; **, p < 0.01; ***, p < 0.001 (gray line: iv-)). (**A**) Created in BioRender. Tenbusch, M. (2025) https://BioRender.com/o92n191.

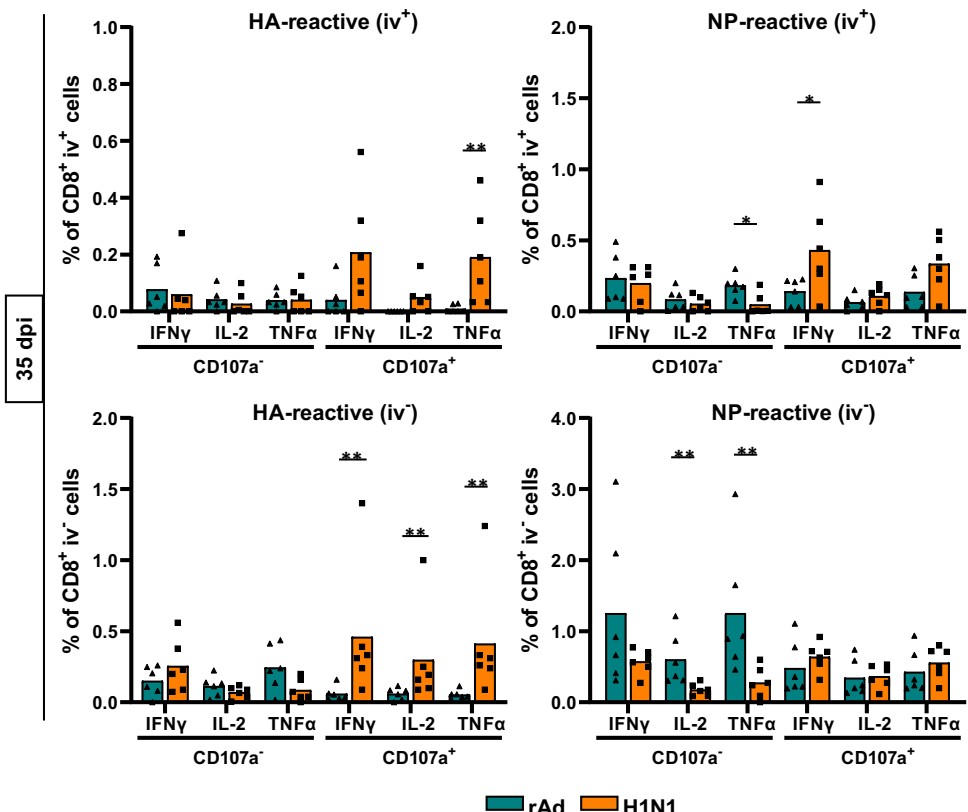

**Fig. 2 | Functional CD8+ T-cell responses measured by ICS.** Lymphocytes from rAd-immunized or H1N1-infected mice were isolated, and ICS was used to identify influenza-reactive T cells. HA- and NP-specific cytokine secretion was measured to determine functional CD8+ subpopulations, which are shown for day 35 as frequencies of the circulating (iv+) or tissue-resident (iv) CD8+ T-cell compartment. Each data point represents an individual mouse, and the bars represent the mean of the group ($n = 6$ mice per group). Statistical significances were analyzed by a two-tailed Mann-Whitney test (*, $p < 0.05$; **, $p < 0.01$).

producing IFNγ, IL-2, or TNFα after stimulation with HA-specific peptides was significantly higher in H1N1-infected mice on day 35 post-infection (Fig. 2), although the absolute number of HA+ Pent+ CD8+ T cells was higher after immunization than after infection at this time point (Fig. 1B). In contrast, cytokine-producing NP-specific $T_{RM}$ were more pronounced in the immunized animals. The most striking difference between the two treatments was the detection of CD107a-expressing subpopulations, marking degranulation. H1N1-induced T cells were predominantly positive for CD107a, whereas the cytokine-producing T cells induced by adenoviral vector immunization were mostly CD107a negative, which might suggest alterations in the cytotoxic capacity of those cells (Fig. 2). This observation was also true 150 days post-treatment, but due to the overall lower cell numbers it did not reach statistical significance anymore (Supplementary Fig. 5).

Next, to CD8+ $T_{RM}$, we analyzed the generation and phenotypic appearance of CD4+ $T_{RM}$ induced by immunization or H1N1 infection. In the absence of suitable MHC-II pentamers, antigen-experienced cells were identified by the expression of CD44, a commonly used marker of T-cell activation[30] (Supplementary Fig. 6). At the peak of the T-cell response, the absolute numbers of CD4+CD44+ T cells, negative for the iv- staining (iv), was significantly higher in vaccinated mice compared to virus-infected mice (Supplementary Fig. 7A). Similarly to mucosal Pent+ CD8+ T cells, the absolute number of CD4+ CD44+ T cells, dropped specifically within the first five weeks after the initial treatment. Thereafter, antigen-experienced CD4+ T cells were maintained throughout the memory phase for at least 150 days. Iv CD4+ CD44+ T cells were classified into different $T_{RM}$ phenotypes based on the expression of CD11a, CD69, and CD103. However, no significant differences were observed between the treatment groups. The majority of the iv-CD44+ T-cell population exhibited a CD11a+CD69-CD103- single-positive phenotype, followed by CD11a+CD69+CD103- double-positive cells (Supplementary Fig. 7B). This phenotypic distribution did not change much throughout the memory phase. In line with the phenotypic analyses, we did not observe substantial differences between the two treatment groups regarding the cytokine production of CD4+ T cells after antigen-specific restimulation with HA- and NP-derived peptides (Supplementary Fig. 7C).

**Temporal persistence of B and T lymphocytes in the lung parenchyma**

Since different kinetics were observed for the persistence of $T_{RM}$ after mucosal immunization and H1N1 infection, we histologically analyzed the localization of $T_{RM}$ and structural changes in the tissue, such as iBALT or RAMD formation. At day 18 post-treatment, the alveolar architecture of the lungs remained intact in both treatment groups, with no signs of tissue damage or fibrosis (Fig. 3A). Although we used an adenoviral vector encoding IL-1β as an adjuvant, immunization did not result in pulmonary inflammation or airway remodeling. In contrast, the airways of mice infected with H1N1 exhibited regions of inflammatory infiltrates predominantly composed of lymphocytes (indicated by arrows), with very few monocytes present. By immunofluorescence, CD4+ and CD8+ T cells were identified throughout the lung tissue on day 18 post-immunization or H1N1 infection, whereas only a very small number of lymphocytes was present in the lung parenchyma of naïve animals (Fig. 3B). However, in lung sections of infected animals, but not of immunized ones, CD8+ T cells were present in close proximity to iBALT structures. These tertiary lymphoid structures were identified by clustering of B220+ B cells and serve also as the primary niche for CD4+ memory

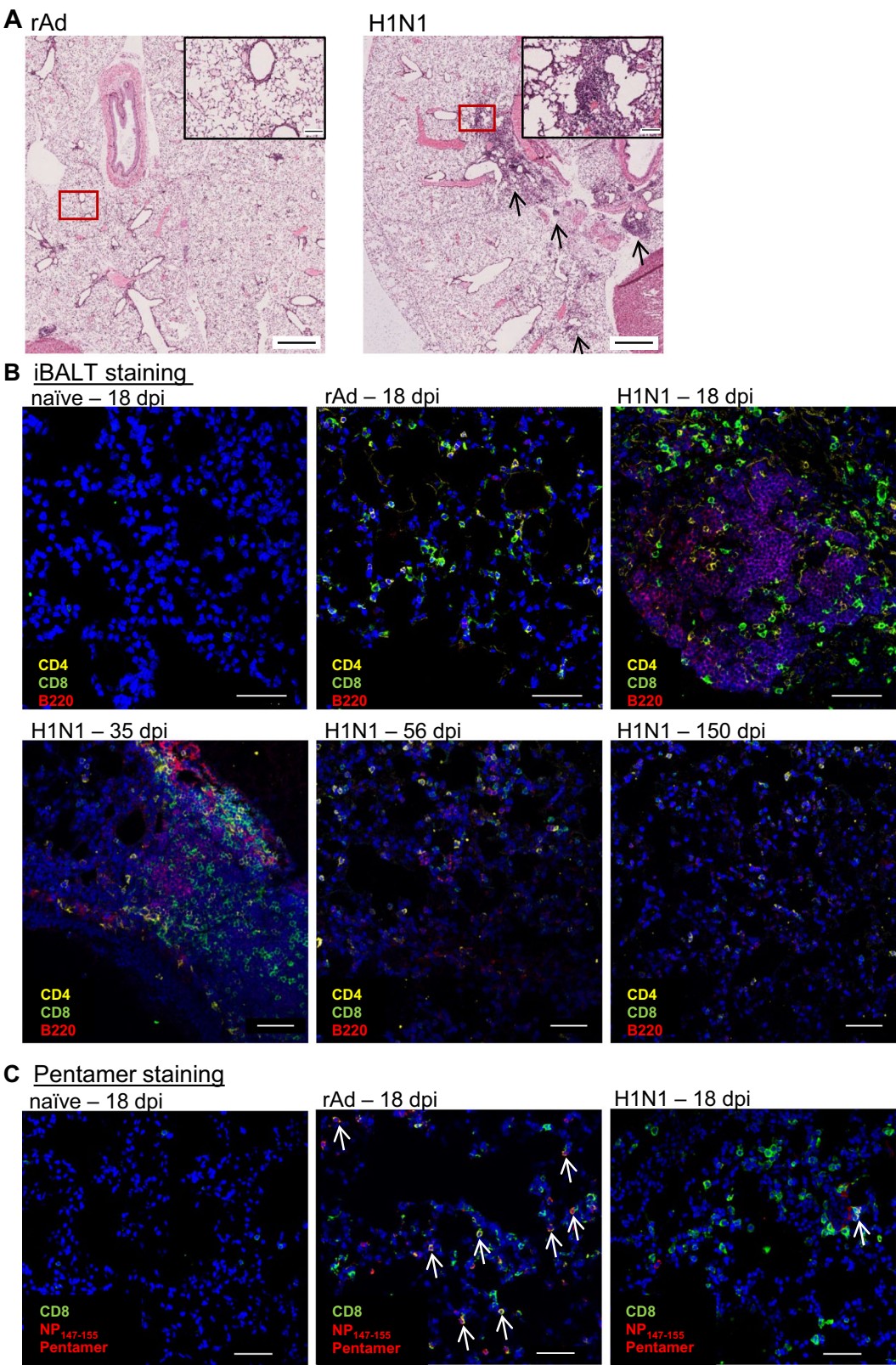

T cells. iBALTs were present as early as 18 days after IAV infection and remained detectable until day 35. Although isolated B cells were still observed 56 days after virus inoculation, the dense B-cell accumulations diminished over time, with only a few immune cells detectable in the late phase of immune memory (>100 days). Such B-cell clusters were absent in sections of vaccinated mice. More specifically, the presence of $NP_{147-155}$-specific $CD8^+$ T cells was confirmed in the lung parenchyma of immunized and infected mice by in situ pentamer staining (Fig. 3C). These antigen-specific $CD8^+$ T cells were not detected in lung sections from naïve animals. However, while the lung tissues of immunized mice showed a high number of $NP_{147-155}$-$Pent^+$ $CD8^+$ T cells (indicated by arrows), most $CD8^+$ T cells in H1N1-infected animals could not be stained with $NP_{147-155}$ pentamers and therefore might have a different antigen-specificity.

**Fig. 3 | Histological analyses of lung tissue post immunization or H1N1 infection. A** Lungs from rAd-immunized or H1N1-infected mice were obtained 18 days post-treatment, and HE-stained lung sections were scanned at 40x magnification. Airways were generally intact and solely H1N1 infection caused lymphocytic infiltration reflected by areas of lymphocyte accumulations, without neutrophils, monocytes, and macrophages (arrows). One representative example of four mice per group is shown. Scale bars in whole slide lung images are 1000 μm and scale bars in magnified images are 100 μm. **B** At the indicated time points, lungs of naïve, immunized, and infected mice were removed, embedded into O.C.T. compound, and finally stained with anti-CD4, anti-CD8, and anti-B220. Nuclei are shown in blue (Hoechst33342), CD4+ T cells are shown in yellow, CD8+ T cells are shown in green, and B cells are shown in red. **C** To identify the localization of antigen-specific CD8+ T cells in the lung parenchyma, lung sections from day 18 p.i. were primary stained with anti-CD8, followed by MHC-I pentamer staining. The signal intensity of bound APC-labeled pentamers was amplified with anti-APC-AF647, and anti-mouse IgG2b-AF647. Nuclei are shown in blue (Hoechst33342), CD8+ T cells are shown in green, and MHC-I pentamer-specific T cells are shown in red. Images were acquired on a Leica SP5X laser scanning confocal microscope using a 40x oil objective and are representatives from a collection of different slices of four mice per group (n = 4). Scale bars in each image are 50 μm.

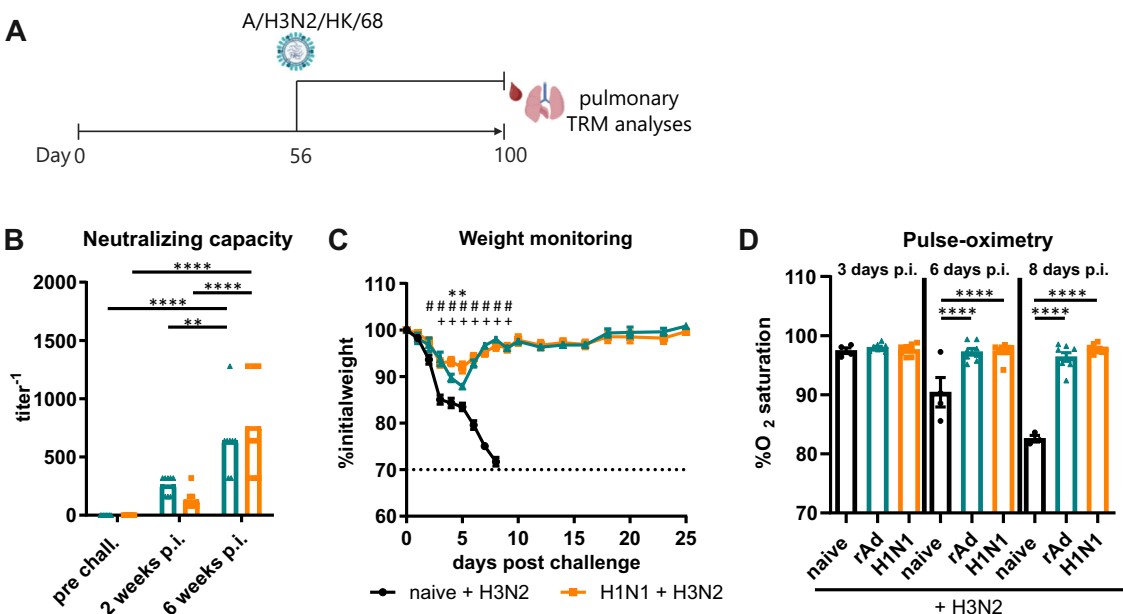

**Fig. 4 | Experimental schedule, neutralizing capacity, bodyweight analysis, and blood oxygen saturation after lethal H3N2 challenge in the mid-memory phase. A** On day 0, seven-weeks-old female BALB/c mice were either i.n. immunized with rAd-HA, rAd-NP, and rAd-IL-1β or infected with the H1N1 strain A/PR/8/34 as described before. 56 days after the initial treatment, mice were challenged with H3N2 A/HK/68 (10,000 PFU) to determine the protective capacity of vaccine- or infection-induced immunity in the mid-phase of immune memory, as well as the fate of pre-existing $T_{RM}$. **B** Serum samples were collected before the challenge as well as two and six weeks after the H3N2 challenge to analyze the neutralizing capacity against heterosubtypic H3N2. Each data point represents an individual mouse, and the bars represent the mean of the group (n = 8 mice per group). **C** Weight loss was monitored daily and is expressed as a percentage of the initial weight on day 0. Naïve mice served as control, but all of them reached the endpoint criteria on day eight after the challenge. Curves represent the mean with SEM (n = 4 mice for naïve, 8 mice for rAd and H1N1). **D** Pulse-oximetry was performed three, six, and eight days p.i., and percentages of blood oxygen saturation are presented. Each data point represents an individual mouse, and bars represent the mean of the group mean ± SEM per group (n = 4 mice for naïve, 8 mice for rAd and H1N1) (**B**). Statistical significances were analyzed by two-way ANOVA followed by Tukey's multiple comparisons test (**B**; **, p < 0.01; ****, p < 0.0001)., by two-way ANOVA followed by Tukey's multiple comparisons test (**C**; *, p < 0.05 H1N1 vs. rAd; #, p < 0.05 H1N1 vs. naïve; +, p < 0.05 rAd vs. naive) or by one-way ANOVA followed by Tukey's multiple comparison test (**D**; ****, p < 0.0001). **A** Created in BioRender. Tenbusch, M. (2025) https://BioRender.com/z57x523.

## Heterosubtypic immunity conferred by primary immunization and H1N1 infection

In the absence of neutralizing antibodies, cross-reactive $T_{RM}$ are capable of providing heterosubtypic immunity and protection from severe IAV-induced disease by rapid, local control of viral replication[7,9,10,26]. In this context, mucosally immunized or H1N1-infected mice were challenged to compare their ability to control an infection with the heterosubtypic H3N2 virus (Fig. 4A). To underline the contribution of the T-cell response, we confirmed first the absence of neutralizing antibodies against the infecting H3N2 strain. Although all treated animals developed substantial amounts of IgG antibodies against the H1N1-derived antigens HA and NP, serum IgGs able to bind the heterosubtypic HA could not be detected in our flow-cytometer based assay (Supplementary Fig. 8). Consequently, no neutralizing antibodies to H3N2 could be detected prior to the challenge and, in all animals, only developed de novo as a consequence of the challenge infection (Fig. 4B). Considering the correlation of viral replication and antibody titers, the comparable titers of H3N2-neutralizing antibodies two and six weeks after the infection indicated a similar degree of viral disease in both groups. This is supported by weight loss as a disease parameter. Both treatment groups reached their minimum weight on day five post-infection before full recovery was achieved within three weeks. In contrast, all mice of the naïve group reached the endpoint criteria on the eighth day after viral challenge (Fig. 4C). Pulse-oximetry conducted on days three, six, and eight confirmed that both the adenoviral vaccine and primary H1N1 infection protected from impaired lung function, which could be seen in naïve animals (Fig. 4D).

Since the focus of our study lays in the differential induction of antigen-specific $T_{RM}$, we were interested in the fate of the pre-existing T cells in response to this secondary challenge. For this purpose, immunized or infected sets of mice were kept for the same time span (100 days post-treatment) without challenge and were then analyzed in parallel with challenged mice for HA- and NP-specific T cells (Fig. 5). Since it has previously been postulated that $T_{RM}$ might undergo

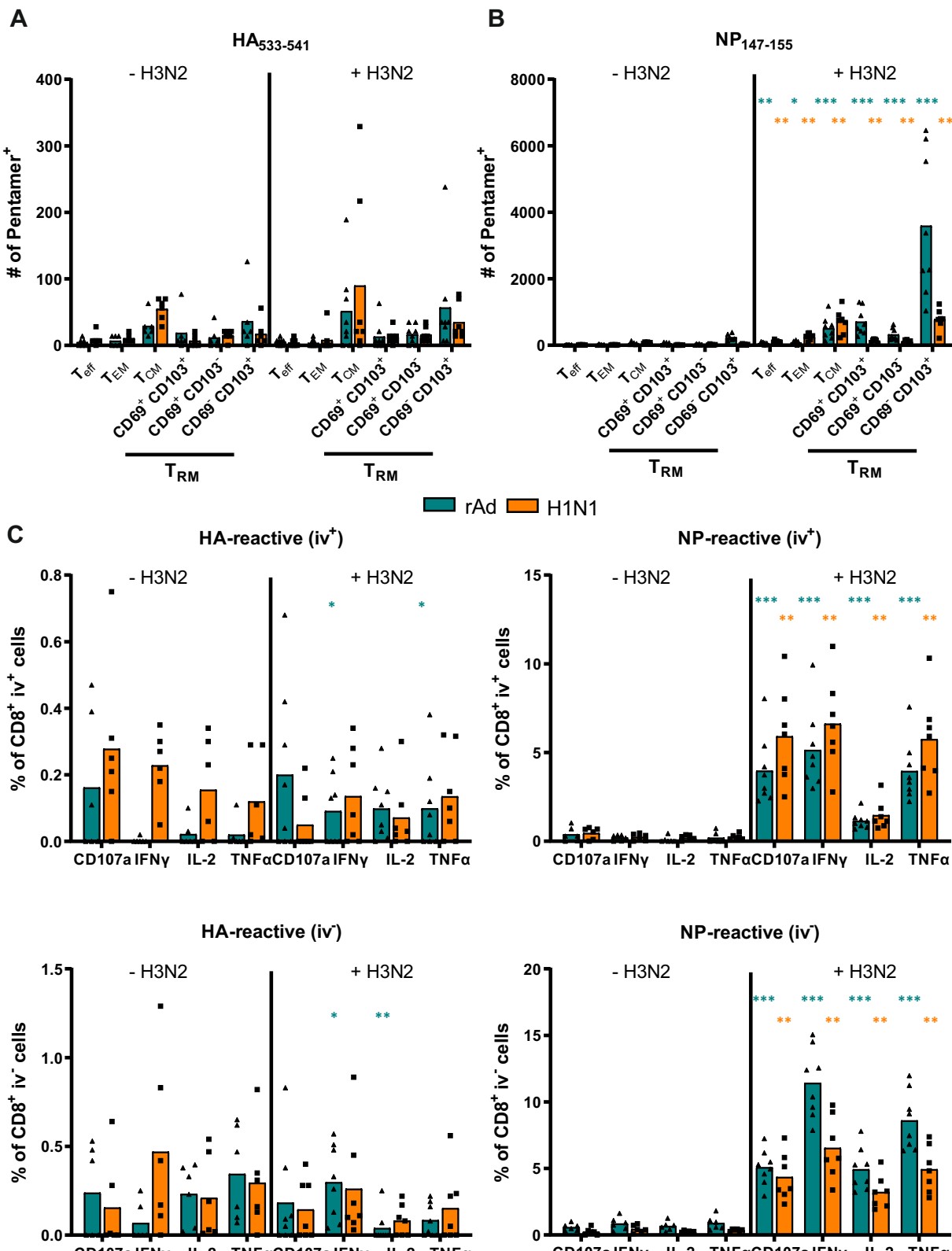

**Fig. 5 | Phenotypic and functional profile of CD8+ T cells after H3N2 challenge.** Previously, rAd-immunized or H1N1-infected mice were challenged with H3N2 (10,000 PFU) 56 days after the initial prime, and lymphocytes from lung tissues were isolated 44 days after the challenge (+ H3N2). One set of mice was not reinfected and served as the control group (- H3N2). Additionally, some lymphocytes were restimulated in vitro using MHC-II- or MHC-I-restricted peptides derived from HA and NP, and ICS was used for the functional identification. **A, B** Phenotypic differentiation between antigen-specific $T_{eff}$, $T_{EM}$, and $T_{CM}$, and different subsets of $T_{RM}$. The graphs show the total number of $HA_{533-541}$- and $NP_{147-155}$-specific Pent+ CD8+ T cells. **C** Frequencies of cytokine-specific CD8+ T cells investigated in secondary infected mice (+ H3N2) compared to only primed mice (- H3N2) are shown. **A–C** Each data point represents an individual mouse, and bars represent the mean of the group ($n = 6$ mice for rAd and H1N1 (-H3N2), 7 mice for H1N1 (+ H3N2) and 8 mice for rAd (+ H3N2)). To compare statistical effects between unchallenged (- H3N2) and challenged (+ H3N2) mice of one group, statistical significances were analyzed by two-tailed Mann-Whitney test (*, $p < 0.05$; **, $p < 0.01$; ***, $p < 0.001$).

apoptosis in case of inflammation and the absence of cognate antigen[31], we analyzed whether $HA_{533-541}$-specific $T_{RM}$ will disappear due to H3N2-induced inflammation because this epitope is not present in H3. However, we neither observed significant alterations in the number of HA-specific memory cells (Fig. 5A) nor in the frequency of cytokine producing CD8$^+$ T cells (Fig. 5C) after the H3N2 challenge, independent from the primary treatment. In contrast, the number of $NP_{147-155}$-specific T cells significantly increased in response to the secondary infection with H3N2, which shares this epitope. Specifically, the numbers of CD103$^+$ $T_{RM}$ were elevated independent of their CD69 status (Fig. 5B). In previously H1N1-infected mice, the expansion of NP-specific $T_{CM}$ and $T_{EM}$ was most obvious, but there was also an increase in the CD69$^-$CD103$^+$ $T_{RM}$ compartment (Fig. 5B). In line with the pentamer data, our functional assay confirmed the elevated numbers of cytokine-producing, NP-specific CD8$^+$ T cells in the challenged animals compared to the non-infected ones (Fig. 5C). Again, the frequencies of the IFNγ- and TNFα-producing $T_{RM}$ were significantly higher in the adenoviral vector immunized and challenged animals, whereas in the systemic compartment, the expansion of cytokine-producing cells was more pronounced in the H1N1-primed animals (Fig. 5C). Interestingly, NP-specific CD4$^+$ $T_{RM}$ responses were boosted by the H3N2 infection only in the immunized mice and not in the H1N1-infected ones, which is indicated by significantly increased numbers of IFNγ- and TNFα-producing CD4$^+$ $T_{RM}$ (Supplementary Fig. 9).

The contribution of cross-reactive T cells to IAV heterosubtypic immunity has been described before, which is here supported by our observation of a massive expansion of NP-specific T cells in response to the secondary infection with H3N2. In fact, the re-activation of those cells in the tissue led to the rapid control of viral replication and thereby improved the disease outcome. Even seven months after the primary immunization or infection, all animals with pre-existing immunity survived a lethal H3N2 infection and had reduced viral loads in BALF and in the lung compared to naïve animals (Supplementary Fig. 10). At this later stage, weight loss was less severe in the H1N1 group compared to the rAd group, but no significant differences were observed in regard to viral RNA copy numbers.

## The fate of pre-existing antigen-specific $T_{RM}$ is unaffected by unrelated secondary virus infection or sterile inflammation

Next, we analyzed the fate of pre-existing lung $T_{RM}$ upon secondary inflammatory events, which are unrelated to the original antigens. First, immune-primed animals were subsequently infected with RSV to address whether IAV-specific $T_{RM}$ were replaced by RSV-specific $T_{RM}$ through resolution of existing and recreation of new memory depots triggered by the infection-induced inflammation. Alternatively, the co-existence of several $T_{RM}$ populations could lead to spatial pathogen-specific clustering in the lung parenchyma. The quantities and cytokine production capability of CD8$^+$ T cells specific for IAV $HA_{533-541}$, $NP_{147-155}$, and RSV $M2_{82-90}$ were assessed three weeks after the secondary infection and compared to control mice that did not receive the challenge (Fig. 6A). Since RSV infection is not highly pathogenic in mice, minor weight loss was observed with significant differences only between previously H1N1-infected mice and naïve mice seven and eight days after challenge (Fig. 6B). Compared to the previous longitudinal study, the adenoviral vector vaccine seemed to be more immunogenic and induced increased numbers of HA- and NP-specific T cells. However, this difference in numbers has no influence on the analysis of the impact of RSV infection on the fate of previously established IAV-specific T cells. Although no cognate antigen was provided, influenza-specific $T_{RM}$ persisted in the lung after RSV infection, and de novo M2-specific $T_{RM}$ were efficiently primed in both groups (Fig. 6C). Interestingly, the number of HA- and NP-specific Pent$^+$ $T_{RM}$ was slightly reduced after the RSV infection in the rAd group compared to those animals which had been only immunized, but this was not reflected in the frequency of functional, cytokine-producing

$T_{RM}$ (Fig. 6C, D). In the H1N1 group, there were no significant alterations in the amount of Pent$^+$ and cytokine-producing $T_{RM}$ upon RSV challenge indicating that there is almost no impact on the fate of pre-existing $T_{RM}$. Immunofluorescence staining of lung sections confirmed the co-existence of CD8$^+$ $T_{RM}$ with distinct pathogen specificity, as we detected $NP_{147-155}$- and $M2_{82-90}$-specific cells in close proximity to each other (Fig. 6E). Besides, some scarcely distributed antigen-specific cells within the lung tissue, these NP and M2 Pent$^+$ cells were primarily found adjacent to bronchus-associated B-cell clusters, but no pathogen-specific clustering in a zonal pattern within the lung parenchyma was observed. Subsequent RSV infection resulted in the development of small B-cell accumulations also in immunized mice, where iBALTs were absent after primary immunization, while such tertiary lymphoid structures were already formed after primary treatment in H1N1-primed mice.

To address the fate of $T_{RM}$ in a second model of lung inflammation, primed mice were challenged intranasally with LPS 56 days post-treatment. As response to the LPS administration, mice experienced a substantial decrease in body weight for the first 48 h, and this inflammatory process was predominantly characterized by a neutrophilic infiltration (Supplementary Fig. 11A, B). Similar to the findings of the RSV challenge, the LPS treatment had almost no impact on the persistence of IAV-specific $T_{RM}$ confirmed again by pentamer staining and ICS (Supplementary Fig. 11C, D). In summary, we did not observe the disappearance of existing $T_{RM}$ in response to various events of secondary inflammation in the absence of cognate antigen recognition.

## Immunization or infection results in different transcriptional profiles of NP-specific $T_{RM}$

Since we detected phenotypic and functional heterogeneity in the $T_{RM}$ compartments, NP-Pent$^+$ CD8$^+$ T cells, either induced by immunization or by H1N1 infection, were sorted by FACS, and scRNA-seq was performed. Almost all cells were positive for *Cd8a* mRNA, and only minor impurities of CD4$^+$ Th or CD19$^+$ B cells could be detected, which confirms the efficient enrichment of CD8$^+$ T cells by the sorting (Supplementary Fig. 12A). According to the Leiden algorithm, the cells were divided into nine clusters, which were all classified as memory CD8$^+$ T cells by automatic cell type annotation (ScType, Fig. 7A). There were four clusters (0, 2, 4, 6) dominantly representing cells from H1N1-infected animals and three major clusters (1, 3, 8) for the immunized group. Interestingly, cells of cluster 3 exhibited a unique transcription profile, which resembled CD8$^+$ T cytotoxic 17 cells (Tc17)[32], marked by high expression levels of *Il17f, Il17a, Rorc, Ccr4, Ccr6* (Fig. 7B and Supplementary Fig. 12B). Unlike Th17, these cells show no *Cd4* expression but are high in *Cd8a* expression. In contrast to all other populations, cluster 3 does not seem to have typical CTL markers like *Gzmb* or *Prf1*, which are also absent in Tc17, suggesting a different functionality compared to cells of the other clusters (Fig. 7B and Supplementary Fig. 12C). Individual mice could be identified via Hashtag antibodies, which revealed that cluster 3 represented almost exclusively cells assigned to one mouse (rAd_3) of the immunized group (Supplementary Fig. 12D). Notably, all other vaccinated mice exhibited nearly uniform gene expression profiles as revealed by DE analysis, whereas the H1N1-infected animals showed generally a higher variability at the transcriptomic level (Supplementary Table 1). In the next step, we compared the expression profiles of the different memory T cell populations. In line with our flow cytometric analyses (Fig. 1), we did not find substantial numbers of effector or effector memory T cells, which would be identified by the expression of *Klrg1* and *Cx3cr1* (Fig. 7B). However, $T_{CM}$ populations could be easily identified by elevated levels of *Ccr7* and *Sell* (encoding for CD62L) and $T_{CM}$ derived from vaccinated or H1N1-infected mice clustered together in cluster 5. The cells of the other clusters show all some characteristics of $T_{RM}$ in regard to transcription factors, cytokine profile, and/or surface receptors (Fig. 7B). Due to this heterogeneity, we refrain from further

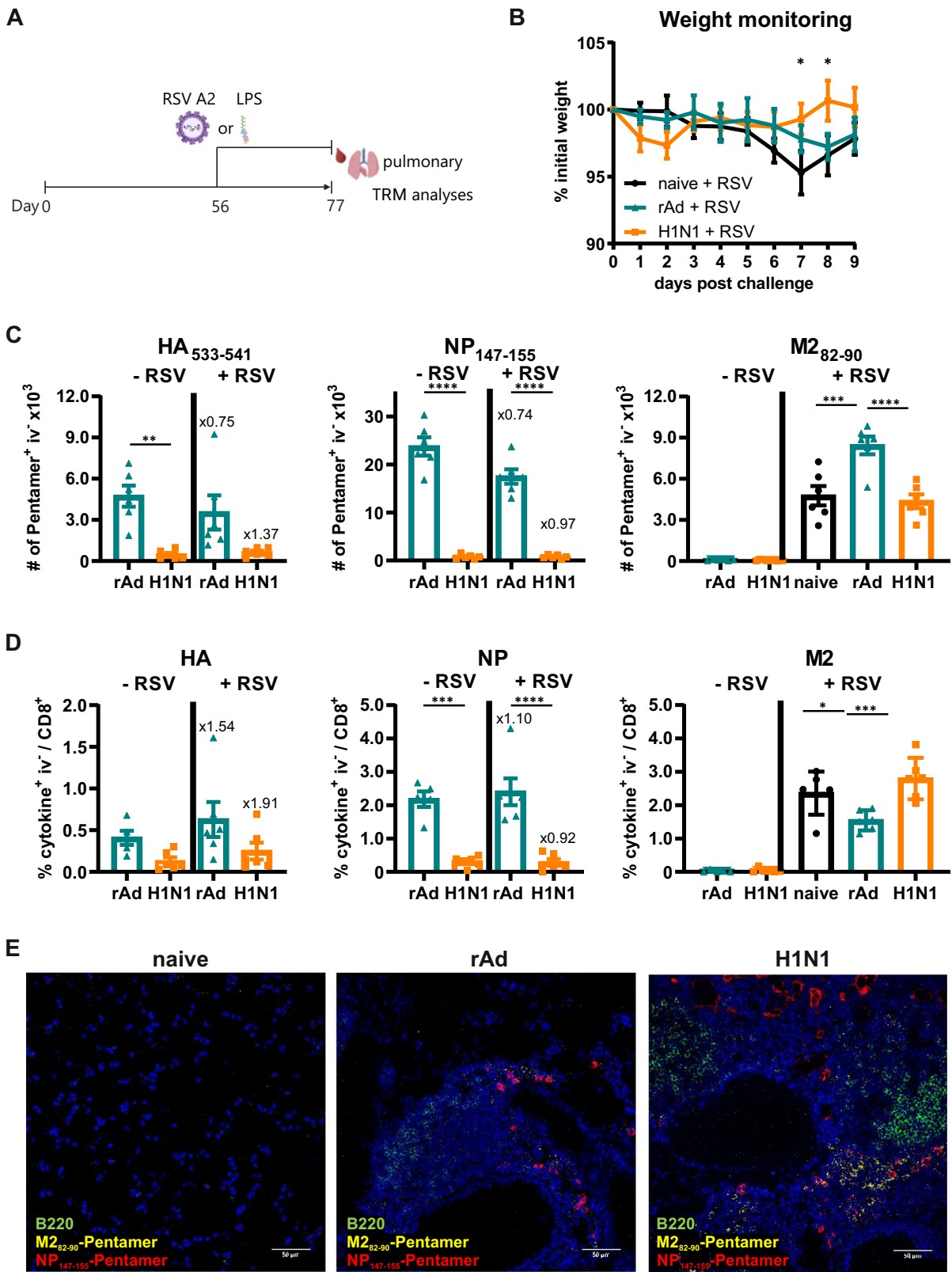

subdividing this population and performed differential gene expression analyses from non-T$_{CM}$ cells of both groups, which will mostly reflect the different T$_{RM}$ population. DE analysis confirmed no significant differences in the T$_{CM}$ populations, but we identified 379 significant differentially expressed genes in the other compartments of the two treatment groups (Fig. 7C and Supplementary Data 1). NP-specific T cells of immunized animals exhibited significantly higher

expression of genes linked to T-cell localization into the lung or T-cell adhesion like *Cxcr6* and *Itgae*. Further, they were enriched for genes induced by interferon signaling (*Stat1, Igtp, Gbp4, Gbp6, Gbp10, Ifit3*) as well as for genes associated with memory T-cell transition, activation, and function (*Ly6a, Cd44, Ctla4, Ahnak*). Although the differential expression did only reach significance for *Havcr2* (encoding for TIM-3) and not for *Pdcd1* (encoding for PD-1), vaccination-induced T$_{RM}$

**Fig. 6 | Experimental schedule to analyze the fate of functional antigen-specific CD8$^+$ T$_{RM}$ upon unrelated secondary virus infection. A** The survival of previously induced IAV-specific T$_{RM}$ was examined upon secondary inflammatory events by subsequent infection with the unrelated RSV ($1 \times 10^6$ PFU) or by LPS treatment ($10\,\mu g$) on day 56 after priming. Three weeks after the challenge, lymphocytes were isolated from lung tissues. One set of initially primed mice remained without secondary exposure (- RSV, - LPS, respectively) and served as the control group. **B** Body weight was measured for nine days following the RSV challenge, and values represent percentages of the initial weight on day 0. **C** Antigen-specific CD8$^+$ T cells were identified by pentamer staining. Total numbers of HA$_{533-541}$-, NP$_{147-155}$-, and M2$_{82-90}$-specific tissue-resident CD8$^+$ T cells are shown. **D** Frequencies of non-circulating CD8$^+$ T cells that are at least positive for the single expression of either CD107a, IFNγ, IL-2, or TNFα were assessed after in vitro restimulation. **E** Immunofluorescence staining of lung tissues from initially rAd-immunized or H1N1-infected mice reinfected with RSV on day 35 and collected an additional 35 days after challenge. Tissue sections were stained with anti-B220 (green), RSV M2$_{82-90}$ pentamer (yellow), and IAV NP$_{147-155}$ pentamer (red). Nuclei are shown in blue (Hoechst33342). Images were acquired on a Leica SP5X laser scanning confocal microscope using a 40x oil objective and are representatives from a collection of different slices of three mice per group ($n = 3$). Scale bars in each image are $50\,\mu m$. **C, D** Each data point represents an individual mouse, and bars represent the mean of the group mean ± SEM per group ($n = 6$ mice per group, only in (D) one sample in the group rAd (-RSV) was lost due to technical issues). Numbers above the columns show fold change between unchallenged (- RSV) and challenged (+ RSV) mice. Statistical significances were analyzed (**B**) by two-way ANOVA followed by Tukey's multiple comparison test, or (**C, D**) by one-way ANOVA followed by Tukey's multiple comparison test ((**B**): *, $p < 0.05$ H1N1 + RSV vs. naïve + RSV; (**C, D**); *, $p < 0.1$; ***, $p < 0.001$; ****, $p < 0.001$). (**A**) *Created in BioRender. Tenbusch, M. (2025)* https://BioRender.com/g83h450.

showed a distinct expression of such exhaustion-related genes as seen specifically in cluster 8 accompanied by the highest level of *Gzmb* transcripts. In contrast, antigen-specific T cells from H1N1-infected mice showed reduced expression levels of *Itgae* but higher levels of *Cd69* (cluster 2 and 4, Fig. 7B). They further exhibited elevated expression levels of genes associated with cytotoxicity and effector functions, such as *Gzmk, Ifng, and Tnf*. In addition, these infection-induced T cells displayed typical memory markers, including *Cd7*, and a resting cell signature, reflected by *Jun, Fosb, Dusp1*, and *Dusp2* expression. Overall, we identified different gene expression profiles of T$_{RM}$ depending on the initial inflammatory conditions, but also core signatures of T$_{RM}$ shared by cells of both groups. These include characteristic T$_{RM}$ transcription factors such as *Znf683* (encoding Hobit) and *Runx2*, which are important for tissue-resident memory differentiation and maintenance[33], as well as *Ccr5* as a crucial chemokine receptor in the early response of memory CD8$^+$ T cells[34] and typical T$_{RM}$ signature genes such as *Itga1* and *Itgae*, even though the expression levels of these genes varies between the clusters. Finally, we analyzed the TCR repertoire from our scRNA-seq data to get an overview about the clonotype network. We identified a total of 5409 cells that had a functional TCR. Focusing on the non-T$_{CM}$ population, we observed considerable clonality in both groups, with Gini indices of 0.790 or 0.762 in the vaccinated or infected animals, respectively. 572 clonotypes found in the T$_{RM}$ compartment showed a clone size up to 349 cells (rAd) or 654 cells (H1N1) (Supplementary Data 2). Of those, 534 clonotypes were found exclusively in the T$_{RM}$ compartment (Gini index: rAd: 0.706; H1N1: 0.682), and 38 were shared between T$_{RM}$ and T$_{CM}$, suggesting a common clonal origin. Due to the specific sampling of lung-derived T cells, the number of T$_{CM}$ is rather small, and almost all of the T$_{CM}$-related clonotypes were singletons (Gini Index: rAd: 0.005; H1N1: 0.000). Three, so-called public clonotypes, were found in the tissue-resident memory T-cell compartment of mice from both treatment groups, but their frequency as a percentage of the total was relatively low (Supplementary Data 3).

## Discussion

Current influenza vaccine programs focus primarily on the induction of strain-specific antibodies directed against the variable surface proteins of the virus, thereby providing protection only against infection with homologous IAV strains. In contrast, natural influenza infection additionally stimulates cross-reactive memory T-cell responses that target highly conserved internal viral antigens and, subsequently, provide moderate levels of HSI[35,36]. Furthermore, mucosal vaccination using adenoviral vectors efficiently induces long-lasting T-cell responses and protects against infection with heterosubtypic IAV strains[23,37]. However, it is not clear whether T$_{RM}$ triggered by immunization or natural infection represents a uniform population. Generally, two competing and non-exclusive models have been proposed how naïve T cells can develop into T$_{RM}$[38]. The model of local divergence supports the idea that multipotent effector T cells will enter the inflamed tissue, where they receive the signals for T$_{RM}$ programming. Other studies suggest already predefined T$_{RM}$ precursor cells among circulating effector cells, which are prone to enter tissues more easily (systemic divergence). Indicated by the different expression of pro- and anti-inflammatory cytokines/chemokines and the different migration patterns of immune cells, we hypothesize that the inflammatory environment in the lung tissue might be the main driver for the T$_{RM}$ programming and, thereby, the origin of different T$_{RM}$ subsets. Although this would be rather in favor of the local divergence model, this formally does not exclude the existence of preconditioned T$_{RM}$ precursor cells in circulation.

Here we analyzed the phenotypic and functional heterogeneity of vaccination- or infection-induced lung T$_{RM}$. Both immune stimulations initially elicited strong antigen-specific T-cell responses, but the distribution of systemic and tissue-resident memory T-cells differed significantly between the treatment groups during the memory phase.

Antigen-specific T$_{RM}$ populations persisted in the lung for more than five months after vaccination, whereas the T$_{RM}$ pool was almost completely lost in influenza-infected mice at that time. This is consistent with the literature, as it has been shown that pulmonary CD8$^+$ T$_{RM}$ induced by adenoviral immunization are maintained in the respiratory tract for at least one year, whereas infection-induced lung T$_{RM}$ are described as relatively short-lived[7,24,39]. The long-term residency of vaccine-induced CD8$^+$ T$_{RM}$ may be attributed to a more pronounced expression of CD103 (*Itgae*) compared to H1N1-induced CD8$^+$ T$_{RM}$, as determined by our flow cytometric and scRNA-seq analyses. While CD69 promotes the accumulation and retention of CD8$^+$ T cells in the lung during the early phase of infection, the maintenance of pathogen-specific memory CTLs in the respiratory tract is more likely due to the adhesive function of CD49a and CD103[11,40]. CD103 is upregulated by TGFβ, a cytokine whose mucosal production is potentiated by IL-1β. Therefore, co-administration of rAd-IL-1β may indirectly enhance the expression of CD103, which mediates adhesion via binding to E-cadherin expressed on lung epithelial cells[41,42]. In addition, CD103 expression was linked to a higher cytotoxic potential of CD8$^+$ T cells in the context of transplant rejections[43] or different tumor models providing a better survival prognosis in the context of cancer[44,45]. In the latter, high CD103 expression was linked to an exhausted phenotype characterized by high expression levels of GzmB and checkpoint inhibitors, such as PD-1 and CTLA-4[46]. Interestingly, we identified such cells in our scRNA analyses (cluster 8) in the vaccinated group, although the lack of CD107a in our functional assays indicate less degranulation. Further studies should be performed to elucidate the role of CD103 expression on long-lived, vaccine-induced lung T$_{RM}$.

The durability of the T$_{RM}$ response could further be affected by differences in the form of antigen presentation by dendritic cells (DCs) during priming or local antigen retention following immunization or infection. Previous studies in mice have shown that adenoviral

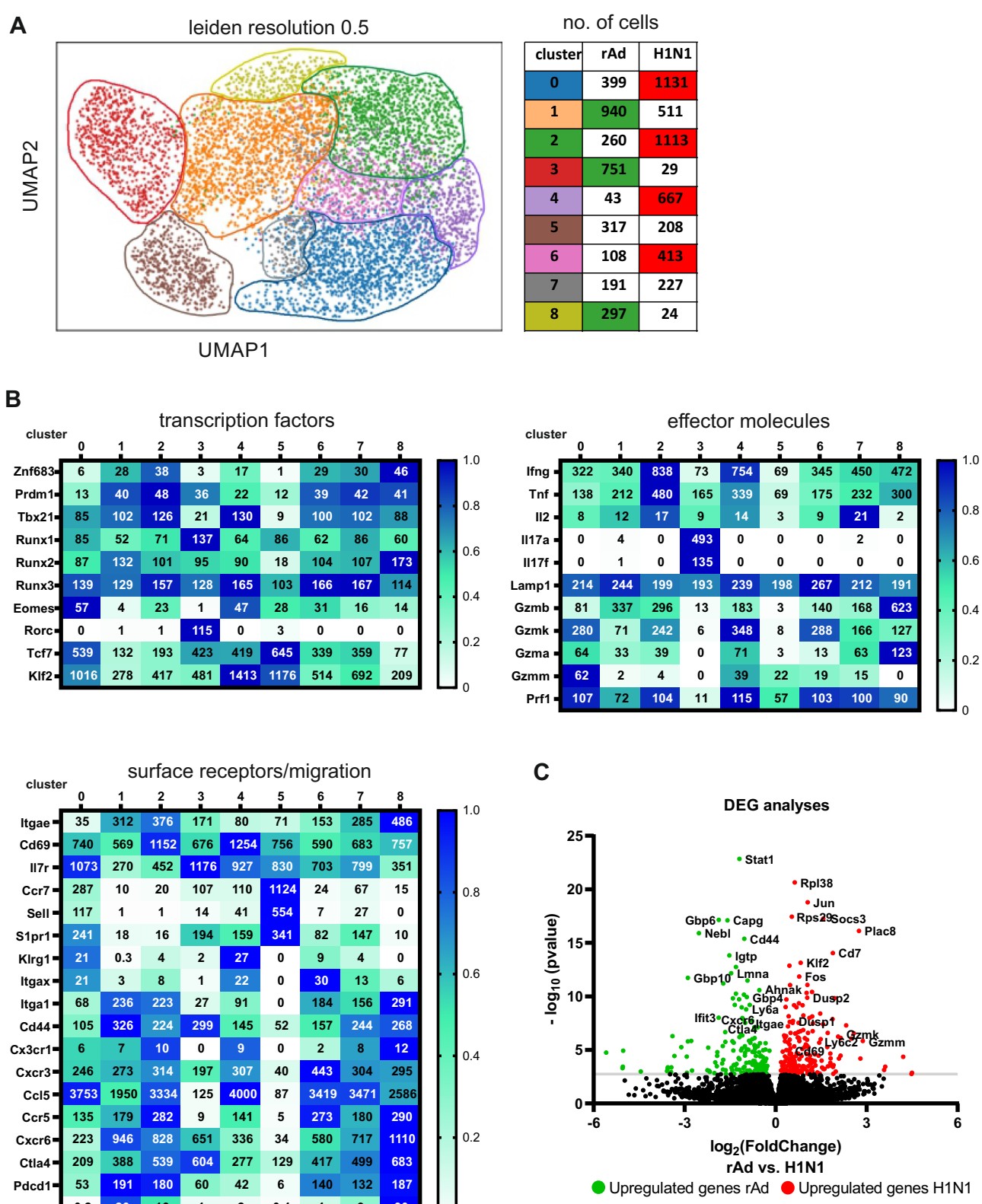

vaccination leads to the persistence of local antigens in the lung parenchyma for at least 110 days. This drives the activation and ongoing proliferation of $T_{RM}$, as well as the continuous recruitment of circulating CD8$^+$ T cells into the $T_{RM}$ pool[24]. Although the persistence of viral antigen depots has also been proposed for the early memory phase after an acute H1N1/PR8 infection, the viral antigen is usually rapidly cleared by a strong adaptive immune response[47]. Considering the

inflammatory conditions in the lung parenchyma upon mucosal vaccination or influenza virus infection, it should be noted that especially the induction of type I IFNs, such as IFNα/β, was significantly different between our two groups. The impact of typ I IFN signaling on the development of CD4$^+$ and CD8$^+$ T-cell memory has been described before[48,49]. However, adjuvanted adenoviral vector immunization induced long-lived $T_{RM}$ without significant induction of type I IFN.

**Fig. 7 | Experimental schedule to define the transcriptional profile of NP-specific T$_{RM}$ after mucosal immunization compared to H1N1 infection.** On day 0, seven-weeks-old female BALB/c mice were either i.n. immunized with rAd-NP and rAd-IL-1β (each $2 \times 10^8$ particles) or infected with the H1N1 strain A/PR/8/34 (100 PFU). On day 56, lymphocytes were isolated from the lung tissue, and NP$_{147-155}$-specific CD8$^+$ T cells were sorted and sequenced by scRNA-seq to determine potential differences between rAd-NP/IL-1β- and H1N1-induced T$_{RM}$ at the transcriptomic level. The analysis includes four mice per group ($n = 4$). **A** Leiden clustering of the dataset with a resolution of 0.5 revealed nine clusters of memory T cells. The clusters are circled separately according to the color code. In the table, the number of cells per cluster is provided for each of the two groups. **B** For selected characteristic genes of memory T cells, the number of transcripts per million and the relative expression levels for each assigned cluster were indicated divided into transcription factors, effector molecules, and surface receptor/migration. The absolute TMP is given as the number within each cell, and the relative expression level is indicated by the color of the cell. For each gene, the TMP of the cluster with the highest expression level was set as maximum, and the ratios were calculated accordingly, ranging from 0 to 1. **C** Volcano plot of a total of 15730 genes analyzed for differential expression. The solid, light gray line indicates the significance threshold. The $p$-value and log2fc was calculated in DESeq2 with default settings. Significant differentially expressed genes are highlighted in green (rAd) or red (H1N1), and selected gene names are indicated.

Here, the early response was characterized by a strong neutrophilic infiltration, which has been also implicated in T-cell recruitment and memory formation in IAV infection models[50]. Nevertheless, further mechanistic studies are needed to decipher the role of those individual contributors to the different inflammatory milieau and T$_{RM}$ programming.

Long-term maintenance of memory Th has also been shown previously[8], and could be facilitated by their localization in local niches, such as iBALTs, in the lower respiratory tract[51]. However, in our study, iBALT formation was observed only after infection and not after mucosal immunization. Within those iBALTs, CD4$^+$ Th directly interacts with memory B cells but also supports the persistence of CD8$^+$ T$_{RM}$ localized outside these lymphoid structures by secretion of IL-21[14,15]. Therefore, the disappearance of infection-induced CD8$^+$ T$_{RM}$ might be linked to a contraction of iBALTs, whereas residency of immunization-induced CD8$^+$ T$_{RM}$ might be based on a different mechanism. While the functional profile of CD4$^+$ T$_{RM}$ was largely comparable after vaccination and influenza infection, functional CD8$^+$ T-cell responses differed between the treatment groups. The majority of vaccine-induced CD8$^+$ T cells did not stain positive for the degranulation marker CD107a, suggesting a reduced ability to release cytotoxic proteins such as granzyme B or perforin after antigen encounter. This might result in less efficient killing of virus-infected cells, but could also limit tissue damage during reinfection. However, vaccine-induced T$_{RM}$ still retains the ability to provide direct effector functions and control viral replication via cytokine release. The secretion of pro-inflammatory cytokines such as IFNγ or TNFα promotes the induction of an antiviral state and further leads to the recruitment of other immune cells to the inflamed tissue[21].

A closer look at the transcriptional profile confirmed the heterogeneity of the differentially induced NP-specific T$_{RM}$. Next to the elevated expression levels of CD103 (*Itgae*) on vaccine-induced CD8$^+$ T cells, their prolonged persistence compared to infection-induced ones could also be attributed to enhanced *Cxcr6* levels. CXCR6 is known to mediate the migration of T$_{RM}$ to the airways but has so far not been implied in the local survival of T$_{RM}$[52]. T cells of the two treatment groups additionally differed significantly in their gene expression of *Ifit3*, which was enhanced after vaccination. The *Ifit3*-encoded protein reduces cell proliferation and increases cellular senescence, but also negatively regulates apoptotic processes[53]. This may reflect the higher number of antigen-specific T$_{RM}$ after immunization when compared to infection during the late memory phase, but at the same time, an enhanced functionality and, consequently, HSI in primarily H1N1-infected mice. Further, T$_{RM}$ formed after vaccination showed increased mRNA levels encoding for inhibitory receptors, such as Ctla4, PD-1, or TIM-3, which could lead to limited reactivation and restricted plasticity of CD103$^+$ T$_{RM}$[29]. The highly cytotoxic and more effector-like T$_{RM}$ phenotype of memory CTLs triggered by influenza infection is reflected by high expression levels of *Gzmk, Gzmm*, and *Jun*, with the latter promoting the expression of effector genes like *Tnf, Ifng*, and *Ccl4*[54,55].

Analyzing the transcriptional profile of lung-localized CD8$^+$ T cells becomes even more complex when considering the various ways in which a gene can regulate T-cell differentiation. For instance, *Jun* is not only an effector marker but is also associated with a transcriptional profile of resting cells, similar to *Fosb, Dusp1*, and *Dusp2*[56,57], which were also detected in IAV-induced non-T$_{CM}$ by DE analysis. Other studies identified *Plac8* as a critical factor for the establishment of memory CD8$^+$ T cells after influenza infection, which was also highly expressed in H1N1-induced T$_{RM}$ in our study, but not in the vaccine-induced T$_{RM}$[58]. This indicates once again the differential cytokine dependency and gene regulation of T$_{RM}$ stimulated by immunization or infection.

In addition to the observation of a higher degree of variability at the transcriptomic level in the T$_{RM}$ population compared to the T$_{CM}$ population, we observed a robust clonal expansion of T$_{RM}$, while all clonotypes found exclusively in the T$_{CM}$ compartment were singletons. The clonal origin of the different memory T-cell subsets has been widely discussed in the past[59–61], but some reports indicate a common naïve T-cell progenitor for both types of memory T cells (local divergence model). In this regard, one study showed that for every abundant T$_{RM}$ clone generated in the periphery, an abundant T$_{CM}$ clone with identical TCR is present in the lymph nodes, suggesting the same clonal origin[62]. However, other studies emphasize that T$_{RM}$ and T$_{CM}$ differ in their lineage, with a significant proportion of T-cell clones giving rise preferentially to tissue-resident or circulating CD8$^+$ T cells[63]. Since our sequencing analysis was primarily focused on T$_{RM}$, it is difficult to draw any conclusion regarding the clonal expansion or a possible common clonal origin between T$_{RM}$ and T$_{CM}$, as the number of lung-localized CD8$^+$ T cells is much higher than the number of circulating CTLs, and we did not analyze the lymph nodes as a primary site of T$_{CM}$.

Although the cellular composition of the memory CD8$^+$ T-cell subsets differed largely in the memory phase, mice from both treatment groups were protected against lethal infection with a heterosubtypic IAV strain. While vaccinated mice still exhibited high proportions of NP-specific CD8$^+$ T$_{RM}$, T$_{CM}$ were the predominant CD8$^+$ T-cell population found in H1N1-infected mice. However, in both groups, the NP-specific CD8$^+$ T-cell pool expanded substantially after the secondary antigen exposure. This indicates that both the local proliferation of T$_{RM}$ and the expansion and conversion of T$_{CM}$ potentially contribute to virus defense[7,64]. In this context, protective immunity against heterosubtypic influenza challenge may be enhanced in H1N1-infected mice by the presence of CTLs with antigen specificity to other conserved IAV epitopes, including PB2, PB1, PA, M1, M2, NS1, or NS2, which were not present in our adenoviral vector vaccine[12,65]. This might also explain the less pronounced weight loss in H1N1-primed mice in response to the H3N2 challenge at seven months post-treatment. In addition, the primary H1N1 infection induced significant higher NP-specific antibody titers than the adenoviral vector immunization. Although the functional mechanism is so far not completely understood, anti-NP antibodies have been shown to be protective in mouse models[66] and might have contributed to viral clearance in our challenge experiments.

Since our main focus in the second part of this study was on the fate of existing T$_{RM}$, we investigated the survival of T$_{RM}$ under inflammatory conditions and in the absence of cognate antigen. Upon

H3N2 infection, we detected the persistence of non-cross-reactive HA-specific CD8[+] T cells, indicating that present pulmonary $T_{RM}$ were not replaced even in the absence of cognate antigen. This was verified by secondary infection with RSV, where both the number of previously induced IAV-specific CD8[+] $T_{RM}$, as well as the frequencies of their functional correlates remained at a constant level upon challenge. Even though we did not observe a clear separation of IAV- or RSV-specific cells in defined areas of regeneration after tissue injury, we could show that pre-existing CD8[+] $T_{RM}$ are not eliminated by the reestablishment of new $T_{RM}$ spots with other pathogen-specificities and are located close to B-cell accumulations. Similarly, LPS treatment had no effect on the survival of formerly induced influenza-specific $T_{RM}$, which contradicts previous findings suggesting that tissue damage in the absence of cognate antigen selectively induces cell death of CD8[+] $T_{RM}$ by the extracellular release of nucleotides during inflammation, which are recognized by P2RX7[31]. The minor effect of the P2X7 receptor on the survival of IAV-specific CTLs in the mucosa could be related to the relative immunological naivety of our mice when compared to, e.g., wildlings or 'dirty mice'[67,68]. Due to more pronounced exposure to microbes and pathogens, such mice may have a larger lung $T_{RM}$ repertoire, and thus selective cell death may be more intended to create new niches for infection-relevant $T_{RM}$ than in our mice that are housed under specific pathogen-free conditions.

Overall, our data highlight that phenotypic and functional characteristics, as well as the maintenance of lung $T_{RM}$ is determined by the inductive stimuli. Mucosal adenoviral vaccination stimulated long-lasting $T_{RM}$ with a less cytotoxic character, whereas local memory cells induced via influenza infection diminished in the late memory phase. Unexpectedly, this was not linked to a reduced subsequent immunity. Thus, we assume that in addition to the number of $T_{RM}$, their phenotypic diversity affects their responsiveness and protective capacity, drawing attention to the functional role of CD103 regarding long-term retention in the lung tissue. Considering the fate of pre-existing $T_{RM}$ during subsequent inflammatory events, we detected a strong boost effect on NP-specific T cells upon secondary antigen encounter, while the absence of cognate antigen did not result in inflammation-induced cell death of former $T_{RM}$. Mucosal vaccination and natural H1N1 infection conferred HSI in mice, indicated by an attenuated morbidity, and reduction in viral load and lung damage. However, the translation of heterosubtypic protection to other species must be handled with caution, as we have seen a different outcome in pigs compared to mice in previous immunization and infection studies, raising potential concerns regarding vaccine development using a single animal model[69]. Given the importance of $T_{RM}$ in controlling viral infections and promoting mucosal immunity, our study highlights the heterogeneity of $T_{RM}$, emphasizing the necessity for a more detailed understanding of $T_{RM}$ for future vaccine development against respiratory viruses.

## Methods

### Ethics statement
The study was approved by the Government of Lower Franconia, which nominated an external ethics committee that authorized the experiments. Studies were performed under the project license AZ 55.2.2-2532-2-1081.

### Adenoviral vector vaccines
The replication-deficient (ΔE1 ΔE3) adenoviral vectors are based on the human serotype Ad5. The encoded sequences for hemagglutinin (rAd-HA) and nucleoprotein (rAd-NP) are derived from H1N1 A/PR/8/34, and rAd-IL-1β contains the sequence for murine mature IL-1β as previously described[23].

### Mice
Five-weeks-old female BALB/cJRj mice were purchased from Janvier (Le Genest-Saint-Isle, France, ref: SC-BALBj-F) and housed in individually ventilated cages in accordance with German law and institutional guidelines under specific pathogen-free (SPF) conditions with constant temperature (20–24 °C) and humidity (45–65%) on a 12 h/12h-light/dark cycle. The housing took place in the animal facility of the University Hospital of Erlangen (Preclinical Experimental Animal Center, PETZ). The studies were conducted according to the guidelines of the Federation of European Laboratory Animal Science Associations (FELASA) and the Society of Laboratory Animal Science (GV-SOLAS). For our study, we used exclusively female mice due to the advantages in regard to the housing conditions. However, since we aim for comparative analyses of different treatment or prevention strategies, the superiority of one strategy above another can be answered independently of the sex of individuals.

### Immunization, viral infections and challenges
For priming, seven-weeks-old female BALB/cJRj mice were either immunized with $2 \times 10^8$ particles of each antigen- and adjuvant-encoding adenoviral vector or infected with a sublethal dose of 100 PFU of influenza H1N1 A/PR/8/34 both in a 50 µl volume via the intra-nasal route. For challenge experiments, mice received either 10,000 PFU influenza H3N2 A/HK/68, $1 \times 10^6$ PFU of the respiratory syncytial virus (RSV), or 10 µg lipopolysaccharide (LPS Merck, Cat: L4524) i.n. in a total volume of 50 µl.

### Weight monitoring
To evaluate the morbidity of viral infections or the effect of LPS treatment, the percentage of body weight loss was determined daily until the mice reached their initial weight again or the pre-defined endpoints. Any mice that lost more than 25% of their initial body weight without gaining weight within the next 48 h were euthanized by an overdose of inhaled isoflurane.

### Pulse-oximetry
Oxygen saturation in blood was measured via a MouseOx™ Pulse-oximeter as an indirect measure of lung function (Starr Life Science, Oakmont, PA). A mouse-adapted pulse-oximeter clip was positioned on the throat of conscious mice, and heart rate, oxygen saturation ($SpO_2$), breath rate, breath distension, and pulse distension were monitored via the MouseOx Plus software.

### Blood sampling and bronchoalveolar lavage fluid (BALF)
For serology, blood samples were collected through puncture of the retro-orbital sinus under light anesthesia. BALF samples were collected at *postmortem* by washing the lungs with $2 \times 1$ ml PBS (Cat: Gibco, 10x, Cat: 70011-036)) through the cannulated trachea.

### Flow cytometric analyses of cellular infiltration
BALF from infected or immunized mice were centrifuged (5 min, $5000 \times g$) and one half of the cellular fraction was stained with anti-Gr1-AF488 (clone RB6-8C5, eBioscience, Cat: 53-5931-82, 1:300), anti-CD49b-PE (clone DX5, eBioscience, Cat: 12-5971-82, 1:300), anti-CD45-PerCP-Cy5.5 (clone 30-F11, BD Biosciences, Cat: 550994, 1:300), anti-CD19-PE-Cy7 (clone 1D3, BD Biosciences, Cat: 552854, 1:1000), anti-F4/80-APC (clone BM8, BioLegend, Cat: 123116, 1:300), anti-CD11b-APC-Cy7 (clone M1/70, BD Biosciences, Cat: 557657, 1:300), anti-CD11c-BV421 (clone HL3, BD Biosciences, Cat: 560521, 1:100), anti-CD4-BV605 (clone RM4-5, BioLegend, Cat: 100547, 1:300), anti-CD8α-BV711 (clone 53-6.7, BioLegend, Cat: 100747, 1:300) and anti-CD3e-BV510 (clone 145-2C11, BioLegend, Cat: 100353, 1:200). All antibodies are listed in Supplementary Table 2.

### Lymphocyte isolation and intracellular cytokine staining (ICS)
To differentiate between tissue-resident and circulating lymphocytes, intravascular (iv)-staining was performed[70]. For this purpose, 2 µg anti-CD45-BV510 (clone 30-F11, BioLegend, Cat: 103138) were injected

intravenously in a total volume of 150 μl PBS three min before euthanizing the animal. After euthanasia, BALF and lungs were harvested for evaluation of the T-cell responses at the indicated time points. First, lungs were cut into small pieces and enzymatically digested in collagenase D (250 μ/ml, Merck, Cat: C7657) and DNase I (80 μ/ml, AppliChem, Cat: A3778) diluted in 2 ml R10 medium (RPMI 1640 (gibco, Cat: 31870-025) supplemented with 10% FCS (Anprotec, Cat: AC-SM-0027), 2 mM L-Glutamine (Gibco, Cat: 35050-038), 10 mM HEPES (Applichem, Cat: A3724), 50 μM β-mercaptoethanol (Gibco, Cat: 31350-010) and 1% penicillin/streptomycin (Gibco, Cat: 15140-122) for 45 min at 37 °C. Disaggregated lung tissues were filtered through 70-μm cell strainers, and red blood cells were lysed by resuspension in ammonium-chloride-potassium (ACK Lysing Buffer, Gibco, Cat: A10492). For in vitro restimulation, 100 μl lung cell suspension was plated in a 96-well plate and 100 μl R10 medium containing monensin (2 μM, Sigma, Cat: M5273), anti-CD28 (1 μg/ml, Invitrogen, Cat: 14-0281), anti-CD107a-FITC (clone 1D4B, BD Bioscience, Cat: 553793, 1:100), and 5 μg/ml of the respective MHC-I/II peptides: MHC-II peptide HA$_{110-120}$ (SFERFEIFPKE), MHC-I peptide HA$_{518-526}$ (IYSTVASSL), or MHC-II peptide NP$_{55-69}$ (RLIQNSLTIERMVL), MHC-I peptide NP$_{147-155}$ (TYQRTRALV), or MHC-I peptide M2$_{82-90}$ (SYIGSINNI) were added and incubated for 6 h at 37 °C. Positive controls were stimulated in an antigen-independent manner by using anti-CD28 (1 μg/ml, Invitrogen, Cat: 14-0282) and anti-CD3ε (2 μg/ml, BD Bioscience, Cat: 553057). Non-stimulated samples were used for subtraction of background cytokine production (negative values were set at zero). After stimulation, the cells were stained extracellularly with anti-CD8α-Pacific Blue (clone 53-6.7, BioLegend, Cat: 100725, 1:300), anti-CD4-PerCP-eFluor710 (clone RM4-5, Invitrogen, Cat: 46-0042-82, 1:2000), and a Fixable Viability Dye-eFluor780 (Thermo Fisher, Cat: 65-0865-14, 1:100). After fixation, permeabilization, and Fc receptors blockage (anti-CD16/-CD32 (2 μg/ml, Invitrogen, Cat: 14-0161-86)), cells were stained intracellularly with anti-IFNγ-PE (clone XMG1.2, BioLegend, Cat: 505808), anti-IL-2-APC (clone JES6-5H4, BioLegend, Cat: 503810), and anti-TNFα-PE-Cy7 (clone MP6-XT22, BioLegend, Cat: 506324) (all 1:300). Data were acquired on an Attune NxT Flow Cytometer and analyzed using FlowJo v.10 software (Tree Star Inc.). The gating strategy is shown in Supplementary Fig. 3 (Supplementary Fig. 3). All antibodies are listed in Supplementary Table 2.

### Flow cytometric analysis of T-cell phenotypes and pentamer staining

Single-cell suspensions were stained with influenza-specific H-2K$^D$ HA$_{533-541}$ pentamer or H-2K$^D$ NP$_{147-155}$ pentamer or RSV-specific H2K$^D$ M2$_{82-90}$ pentamer (all ProImmune, 1:40) for 20 min at 4 °C. Cells intended for CD4$^+$ T-cell analysis were incubated without pentamer staining. In a second extracellular staining step, cells provided for CD8$^+$ T-cell analysis were incubated with anti-CD127-FITC (clone A7R34, BioLegend, Cat: 135008, 1:300), anti-CD103-BV605 (clone 2E7, BioLegend, Cat: 121433, 1:200), anti-CD69-PerCP/Cy5.5 (clone H1.2F3, BioLegend, Cat: 104522, 1:200), anti-CD45.2-PE/Dazzle594 (clone 104, BioLegend, Cat: 109846, 1:500), anti-CD8α-Pacific Blue (clone 53-6.7, BioLegend, Cat: 100725, 1:300), anti-KLRG1-PE-Cy7 (clone 2F1, invitrogen, Cat: 25-5893-82, 1:300), anti-P2X7R-PE (clone 1F11, BioLegend, Cat: 148704, 1:300), and anti-IFITM3-Biotin (clone aa2-57, R&D Systems, Cat: BAF3377, 1:300). Phenotypic characteristics of CD4$^+$ T cells were specified by staining with anti-CD44-APC (clone IM7, BioLegend, Cat: 103018, 1:200), anti-CD4-AF488 (clone GK1.5, BioLegend, Cat: 100423, 1:200), anti-CD103-BV605 (clone 2E7, BioLegend, Cat: 121433, 1:200), anti-CD69-PerCP/Cy5.5 (clone H1.2F3, BioLegend, Cat: 104522, 1:200), anti-CD45.2-PE/Dazzle594 (clone 104, BioLegend, Cat: 109846, 1:500), anti-CD11a-eFluor710 (clone M17/4, invitrogen, Cat: 48-0111-82, 1:300), anti-CXCR3-APC-Fire750 (clone CXCR3-173, BioLegend, Cat: 126539, 1:200), anti-P2X7R-PE (clone 1F11, BioLegend, Cat: 148704, 1:300), and anti-IFITM3-Biotin (clone aa2-57, R&D Systems, Cat:

BAF3377, 1:300). Subsequently, streptavidin-BV711 (BioLegend, Cat: 405241, 1:300) was used for flow cytometric detection of IFITM3. All samples were aquired on an Attune NxT Flow Cytometer, and flow cytometry data were analyzed using FlowJo v.10 software. Supplementary Fig. 4 (CD8) and Supplementary Fig. 6 (CD4) represent the respective gating strategy (Suppl. Figs. 4, 6). All antibodies are listed in Supplementary Table 2.

### FACS-based antibody analysis

Doxycycline-inducible HEK293A cell lines stably transduced with lentiviral particles encoding either HA or NP of H1N1/PR/8/34 or HA of H3N2 A/HK/68 were used for the detection and quantification of specific antibodies in serum and BALF samples. The respective antigen was overexpressed by doxycycline (Sigma, Cat: D9891) stimulation (HA: 100 ng/ml, NP: 400 ng/ml) for 24 h. Following, $1 \times 10^5$ cells/well were seeded in a 96-well round-bottom plate, and HEK293A-NP cells were fixed and permeabilized before antibody staining. For binding of the surface antigen HA, cells were incubated with sera (1:200) or BALF (1:20) diluted in FACS buffer (PBS with 0.5% BSA (Merck, Cat: A4503)) and 1 mM sodium azide (Sigma, Cat: 71289). To bind intracellular NP, samples were diluted in permeabilization buffer (FACS buffer supplemented with 0.5% saponin (Sigma, Cat: 47036-250G-F). Specifically bound antibodies were detected using the polyclonal anti-mouse IgG-FITC (poly4060, BioLegend, Cat: 406001, 1:300) detection antibody or with an antibody mixture of anti-mouse IgA-FITC (polyclonal, Fortis Life Sciences, Cat: A90-103F), anti-mouse IgG1-APC (clone RMG1-1, BioLegend, Cat: 406610), and anti-mouse IgG2a-PerCP-eFluor710 (clone m2a-15F8, invitrogen, Cat: m2a-15F8) (all 1:300). The median fluorescence intensity (MFI) of each fluorophore was measured on an Attune NxT Flow Cytometer and analyzed using FlowLogic™ (Inivai). The exact concentration of each antibody subtype was extrapolated using a standard serum with known antibody concentration. All antibodies are listed in Supplementary Table 2.

### Influenza microneutralization assay

Neutralizing antibodies in sera and BALF were determined in a microplate neutralization assay. Briefly, the samples were incubated with 2000 PFU of influenza H1N1 A/PR/8/34 or H3N2 A/HK/68 for 1 h at 37 °C before adding the mix to confluent MDCK-II cells in a 96-well plates. On day 4 after infection, the medium was removed, and plaques were identified by crystal violet (AppliChem, Cat: 131762.1608) staining. The highest sample dilution, which completely inhibited an infection, determines the neutralization titer. This titer is given as the reciprocal dilution level.

### Quantitative reverse-transcription real-time PCR (qRT-PCR) for viral RNA detection

Viral RNA was isolated from lung and BALF samples using the Nucleo Spin RNA Virus kit (Machery-Nagel, Cat: 740956.250) according to the manufacturer's instructions. Samples were quantified by qRT-PCR (Go Taq 1-Step RT-qPCR kit, Promega, Cat: A6020) for the IAV M gene using the 7500 Real-Time PCR System (Applied Biosystems) and 7500 software v2.3. For M viral RNA analysis, the following primers were used:

for 5′-AGATGAGTCTTCTAACCGAGGTCG-3′, rev 5′-TGCAAAAA-CATCTTCAAGTCTCTG-3′, and rev 5′-TGCAAAGACATCTTCCAGTC TCTG-3′. Results were expressed as absolute RNA copy numbers calculated according a standard curve of RNA preparations with known copy numbers.

### Immunofluorescence microscopy

Mice were euthanized, the trachea was cannulated, and 1 ml of 20% sucrose (Roth, Cat: 9097.1) in PBS and O.C.T. compound (Sakura Finetek U.S.A., Inc., Torrance, CA, Cat: 4583) (1:2 mixture) was injected to prevent the collapse of the pulmonary airways. The trachea was tied with a string. Lungs, trachea, and heart were removed and submerged

into 20% sucrose/PBS over night at 4 °C. Following, the lungs were washed in sodium chloride and snap frozen using liquid nitrogen. 15 µm thick cryosections were prepared at the Institute of Pathology of the University Hospital Erlangen and stored at − 80 °C until use. To get a general overview about the tissue architecture and immune cell infiltration, lung tissue was stained with hematoxylin and eosin (HE). Whole slide images of HE-stained lung sections were scanned using an S210 digital slide scanner (Hamamatsu) and digitally analyzed with QuPath version 0.4.2, open-source software for digital pathology, and whole slide image analysis[71]. Figures were created using ImageJ software. Histopathological changes were confirmed by a pathologist from the University Hospital Erlangen. The precise localization of various immune cells was determined by immunofluorescence staining. Prior to primary staining, the lung sections were fixed in ice-cold acetone/methanol (Acetone: AppliChem, Cat: 211007.1212 / Methanol: Merck, Cat: 1.06008.6025) (1:1 mixture) for 5 min, air-dried, and tissue sections were surrounded using a PAP pen (Science Services, Cat: PAP-Pen Mini). Next, the sections were rehydrated and blocked with 5% FBS in PBS-T (0.05% Tween, Sigma, Cat: P7949) solution, containing anti-CD16/-CD32 (clone 93, invitrogen, Cat: 14-0161-86, 10 µg/ml). For detection of different immune cell specificities, lung sections were stained with CD4-AF488 (clone GK1.5, BioLegend, Cat: 100423, 5 µg/ml), anti-CD8α-AF647 (clone 53-6.7, BioLegend, Cat: 100727, 5 µg/ml), and anti-B220-BV711 (clone RA3-6B2, BioLegend, Cat: 103255, 5 µg/ml) overnight at 4 °C. Nuclei were stained by mounting with ProLong Glass antifade, which contains NucBlue (Hoechst 33342; Thermo Fisher Scientific, Cat: P36981). For the identification of pentamer-specific cells, lung sections were initially stained with anti-CD8α-AF488 (clone 53-6.7, BD Biosciences, Cat: 557668, 5 µg/ml) overnight at 4 °C, before the APC-conjugated Pro5 MHC $HA_{533-541}$ or $NP_{147-155}$ Pentamer (both ProImmune, 1:20) was added overnight at 4 °C. APC fluorescence intensity was amplified by anti-APC-AF647 (clone 936809, R&D Systems, Cat: FAB8927R, 10 µg/ml) and anti-mouse-IgG2b-AF647 (polyclonal, Invitrogen, Cat: A21242, 30 µg/ml) (each 1 h at 4 °C). Stained sections were mounted as described before. To stain and localize antigen-specific cells of various virus specificities, lung sections were stained with anti-B220-AF488 (clone RA3-6B2, BioLegend, Cat: 103225, 5 µg/ml) overnight at 4 °C. Biotin-conjugated Pro5 MHC $RSV-M_{80-92}$ Pentamer (ProImmune, 1:20) was pre-incubated with streptavidin-PE (Miltenyi, Cat: 130-106-790, 1:2000) for 30 min at 4 °C. Similarly, APC-conjugated Pro5 MHC $NP_{147-155}$ Pentamer (ProImmune, 1:20) was pre-incubated with anti-APC-AF647 (clone 936809, R&D Systems, Cat: FAB8927R, 10 µg/ml). Both complexes were mixed together, added to the tissue slice, and incubated 24 h at 4 °C. Finally, fluorescence intensity for the detection of NP-specific cells was intensified by incubation with anti-mouse-IgG2b-AF647 (polyclonal, Invitrogen, Cat: A21242, 30 µg/ml) for 1 h at 4 °C. Stained lung sections were mounted with ProLong Glass antifade including NucBlue (see above) and images were collected using a Leica SP5X laser scanning confocal microscope (Leica Camera AG, Wetzlar, Germany) with a 40x oil objective (PL APO, NA1.75). For image acquisition, LAS AF software was used and image processing was performed with ImageJ software. All antibodies are listed in Supplementary Table 2.

## Cell labeling and sorting
Single-cell suspensions of lung tissues of rAd-NP/IL-1β or H1N1 A/PR/8/34 treated mice were prepared as described above. Cells of each sample were counted (Automated Cell Counter Luna, Logos Biosystems) and labeled with TotalSeq-C anti-mouse hashtag antibodies (TotalSeq-C0301 anti-mouse Hashtag 1 Antibody (Cat: 155861), TotalSeq-C0302 anti-mouse Hashtag 2 Antibody (Cat: 155863), TotalSeq-C0303 anti-mouse Hashtag 3 Antibody (Cat: 155865), TotalSeq-C0304 anti-mouse Hashtag 4 Antibody (Cat: 155867), BioLegend; 0.5 µg for 2x10$^6$ cells) for multiplex single-cell sequencing analysis and stained with APC-labeled H-2K$^D$ $NP_{147-155}$ Pentamer

(ProImmune, 1:40) for 30 min at 4 °C. Afterwards, cells from one group of mice were pooled together and stained with anti-CD8α-BV421 (clone 53-6.7, BioLegend, Cat: 100737, 1:300) and anti-CD45.2-PE/Dazzle594 (clone 104, BioLegend, Cat: 109846, 1:300) for 20 min at 4 °C (Supplementary Table 2). After resuspension in FACS buffer supplemented with 0.05 M EDTA (Lonza, Cat: 51201), $NP_{147-155}$-specific CD8$^+$ T cells were sorted at a MoFlo Astrios Cell Sorter (Beckmann Colter) in the core unit for cell sorting and immunomonitoring of the Friedrich-Alexander-Universität Erlangen-Nürnberg.

## 10x Genomics library preparation and sequencing
For each group, $2 \times 10^5$ barcoded $NP_{147-155}$-specific CD8$^+$ T cells were sorted in the FACS buffer. Samples were centrifuged, resuspended in nuclease-free water, and a master mix containing RT Reagent B, Poly-dt RT Primer, and RT Enzyme C was added This master mix and cell suspension mixture was loaded into Chromium Chip K (10x Genomics) and partitioned into Gel Beads In-Emulsions (GEMs) in a chromium controller (10x Genomics). Single-cell RNA libraries, as well as those for scTCR-sequencing, were prepared according to the Chromium Next GEM Chip K Single Cell Kit (Cat: PN-1000286), the Chromium Next GEM Single-Cell 5′ Kit v2 Kit (Cat: PN-1000263), the Dual Index Kit TT Set A (Cat: PN-1000215), the 5′ Feature Barcode Kit (Cat: PN-1000541), and the Chromium Single-Cell Mouse TCR Amplification Kit (Cat: PN-1000255) (all 10x Genomics, CA, USA). The single-cell RNA-sequencing (scRNA-seq) and VDJ libraries were sequenced by Novogene UK.

## scRNA-seq data processing
Cell-gene matrices were generated via Cell Ranger v6.1.2 (10x Genomics), and the scRNA-seq reads were aligned to the mm10 genome (UCSC, CA, USA) using Cell Ranger count. Cell Ranger v6.1.2 multi-pipeline was applied for scRNA-seq and VDJ-seq analysis. The generated count matrix was loaded into Scanpy v1.9.3[72]. Doublets were removed with HashSolo, invoked via Scanpy[73]. Cells with a mitochondrial content higher than 10% were filtered out. Raw counts were normalized using size factors calculated by DESeq2 v1.34.0[74] and subsequently scaled with Scanpy's log1p function. Data dimensions were reduced using the UMAP algorithm[75]. Based upon the location of the cells on the two-dimensional UMAP plot, the cells were annotated as $T_{RM}$ ($Cd69^{+/-}$ $Itgae^{+/-}$) and $T_{CM}$ ($Ccr7^+$ $Sell^+$). Scirpy v0.11.2 was employed to analyze the T-cell receptor (TCR) data[76]. Calculation of transcripts per million (TPM), as well as selection of cell populations, was performed using scSELpy v1.1.9[77]. For the differential expression (DE) analysis, the data was treated as pseudo-bulk. Per annotation and sample, the raw Unique Molecular Identifier (UMI) count of all cells for each gene was summed. At this point, the data matrix looks like a bulk-seq UMI matrix. The pseudo-bulk matrix was passed to DESeq2 in order to calculate the log$_2$ fold change and the adjusted $p$-value. In order to compare differential expression between single individuals of the same treatment group, we employed MAST (Model-based Analysis of Single Cell Transcriptomics)[78] on the normalized matrix that was also used for dimension reduction with UMAP.

## Statistical analysis
Statistical analyses were performed using GraphPad Prism (version 9.5.1, GraphPad Software, Inc.). In addition, the volcano plot and the heatmap were created via GraphPad Prism.

## Reporting summary
Further information on research design is available in the Nature Portfolio Reporting Summary linked to this article.

## Data availability
The sequencing dataset have been deposited at Gene Expression Omnibus (GEO) under the series accession number GSE261708. All data are included in the Supplementary Information or available from

the authors, as are unique reagents used in this Article. The raw numbers for charts and graphs are available in the Source Data file whenever possible. Source data are provided in this paper.

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

## Acknowledgements

This work was funded from the Interdisciplinary Center for Clinical Research (IZKF) at the University Hospital of the Friedrich-Alexander-Universität of Erlangen-Nürnberg (Advanced Projects A90 and A101; Junior Project 100). Further support was received by the Deutsche Forschungsgemeinschaft (DFG) through the project TE 1001/4-1 and the research training group RTG 2504 (project 761 number: 401821119, to A.V.A, V.V., K.S. and M.T.). The study was further supported by the Bavarian State Ministry for Science and the Arts via the For-COVID project. K.S. is supported by the German Federal Ministry of Education and Research (BMBF, projects 01KI2013, MH/870 and 031L0290B), the Else Kröner-Fresenius-Stiftung (project 2020_EKEA.127) and the DFG through the Heisenberg program (project SCHO 1949/1-1) as well as the Yellow4FLAVI project, funded by the European Union, under the Horizon Europe Program, Grant Agreement 101137459. We are grateful to the animal staff at the Preclinical Experimental Animal Center (PETZ; University Hospital Erlangen, Friedrich-Alexander-Universität of Erlangen-Nürnberg) for providing excellent animal care. We further thank the Institute of Pathology and Comprehensive Cancer Center Erlangen-EMN (CCC) (University Hospital Erlangen, Friedrich-Alexander-Universität Erlangen-Nürnberg), the core unit for cell sorting and immunomonitoring (Friedrich-Alexander-Universität Erlangen-Nürnberg), as well as the working group of K. Schober of the Microbiology Institute (University Hospital Erlangen) for technical assistance.

## Author contributions

A.S., D.L., and M.T. conceived and designed the study. A.S., J.F., J.H., A.V.A., F.O., V.V., P.I., C.S., N.L., U.A., D.L., and M.T. performed the experiments and collected the data. A.S. and M.T. acquired, analyzed and interpreted the data. A.S., M.D., K.K., S.Z., and K.S. performed and supervised the single-cell sequencing analysis. RR and CG validated the

histopathological analyses and images. M.T. obtained the funding. AS and MT drafted the manuscript, which was then critically reviewed and approved by all co-authors.

## Funding

## Competing interests
The authors declare no competing interests.
