## [Transparent Peer Review file · Nature Communications]

Inflammatory conditions shape phenotypic and functional characteristics of lung-resident memory T cells in mice

Corresponding Author: Professor Matthias Tenbusch

Version 0:

Reviewer comments:

Reviewer #1

(Remarks to the Author)

This manuscript reports the results of studies comparing recombinant adenovirus vector vaccination (rAV) to influenza infection for the priming and development of tissue-resident memory T cells. Longevity of the T cells and protective capacity were measured. They further explore the effects on specific activation and non-specific inflammation on the prevalence and persistence of the virus-specific T cells in the lung tissue. Priming with rAV or H1N1 influenza led to distinct T cell phenotypes and gene expression profiles. Of particular note, are the observations that non-specific inflammation in the forms of LPS or heterologous influenza infection do not significantly impact the presence of TRM, while antigen specific activation alters the phenotypes and functions of the responding T cells. In general, the experiments appear to be well controlled and of good quality. The data is carefully presented. There are a few issues with the readability of the figures however.

The authors discuss non-lymphoid tissue sites, "RAMD" in particular, and local tissue sites as a primary niches and mechanisms for generating TRM. However, there are at least two competing and non-exclusive proposed mechanisms for generation of TRM, one systemic and one tissue localized. These are probably complimentary mechanisms and should be included in the discussion. This does not minimize the significance of the tissue sites.

In many of the figures, the axis labels and much other text is too small to be easily readable. Even enlarged on screen, the figures are hard to read (for example distinguishing a "+" from a "-" superscript indicating positive and negative cells and other super- and sub-script text). There is plenty of room in the figures to enlarge the text.

In figure 1, the total numbers and proportions of HA and NP pentamer+ CD8 T cells are depicted in B and C. The reviewer appreciates the reference to the explicit surface phenotypes of the tissue subsets defined by CD69 and CD103 in C. The use of KLRG1 and CD127 is less explicit. The reader must go to the supplemental figures and legends to find it. Can the authors more clearly define the Teff, Tem and Tcm, perhaps including in the figure legend how they used CD127 and KLRG1? Among the Teff, Tem, Tcm, are any of those expressing CD103 or CD69? Likewise, among the CD69 and CD103 single positives, were any of those expressing Teff, Tem, or Tcm phenotypes? Also, if I understand correctly, B presents the total number of pentamer+ cells, but the numbers below the pie charts in C also indicate total HA and NP specific cells. Those numbers are quite different, so why is that? Some of these comments may also apply to figure 5.

The authors mention there are increase numbers of pentamer+ CD8 T cells in the rAd versus the H1N1 infected mice (line 344-347), yet there is no data shown that supports this claim.

Does rAd result in more memory cells systemically or just more resident cells in the lung? Getting the numbers from the mediastinal lymph node and spleen would clear this up.

The authors were initially careful to define the T cells by their surface expression patterns, rather than just applying labels. When discussing the CD4 T cell data, the authors use iv labeling, CD44, CD69, and CD103 to phenotype the cells. Beginning on line 358, the term "tissue-resident" is used interchangeably with iv-negative status. Being iv-negative does not define tissue residency, only that the cells are not in proximity to the vasculature. On line 363, the cells are referred to as "non-circulating". There are no doubt some circulating memory cells as well as resident cells among the iv-negative population. Why not just say iv-negative, CD44+, CD4+ T cells and leave out the pejorative residency adjectives?

Residency can only be demonstrated in parabiosis type experiments. Accurate terminology is important.

CD103 is the most commonly used marker of CD8 TRM, but it is not a good marker of CD4 TRM. Some have said CD4 TRM do not express CD103 at all. The data in the paper support that as most of the iv-neg (aka "tissue resident" in this paper) are CD103-. While the results of the CD4 analysis are a minor aspect of this work, it is still less ambiguous to refer to the cells populations by their specific phenotypes rather than applying interpretive labels.

Can the authors better define how they determined what was a bronchiole? The images in figure 3 indicate the location of so-called bronchioles, but some look no different than the non-bronchiolar tissue. I trust their analysis did include some specific criteria, but that isn't mentioned.

Figure 6D please label the HA, NP, and M peptides consistently with 6C labeling. It was confusing.

Line 465, this may be semantics, but when discussing the numerical differences between the earlier and later longitudinal studies, "stronger" is not as precise as saying "increased numbers of". Stronger can represent both number and effector function.

Regarding the discussion of the scRNAseq results. It was noted that a putative Tc17 population was identified as cluster 3. In line 505, a sentence starts out saying "as the only TRM population..." Is that a typo? Why is cluster 3 the only TRM population? (even if it's from only the one odd mouse)

Figure 5: The authors classify NP pentamer-specific CD8+ T cells within their scRNA-seq dataset as either corresponding to Tcm or Trm populations, yet it is highly likely that there are T cells that are not either Tcm or Trm in the lung as well, in part based on the data represented in earlier figures. The annotation would be more convincing if the authors would show 1) expression of selected memory subset marker genes overlaid on the shown UMAP plots and 2) heatmaps with per-cluster differential gene expression for the selected marker genes. Additionally, there are several treatment-specific clusters (e.g. C0, C3, C8) which are minimally discussed elsewhere in the text. The authors should consider reanalyzing these data to perform cluster-specific annotations of populations that may differ between treatment groups – these analyses could also include specific visualizations of transcript expression for the cytokines profiled in Figs. 2 & 5.

The color scheme for the heatmap depicted in figure 7E is not conducive to visualizing the weaker positive expression levels for some genes. Itgae is a good example. Perhaps it's just my eyes but it's a challenge to distinguish slight differences in shades of light blue. Were other color schemes tried? Red to green is often used, but obviously isn't great for those who are colorblind.

In the discussion, the authors allude to a "previously unappreciated functional role for CD103" but fail to speculate on what that might be. Would they care to speculate? For example, in 2004 Donna Farber showed that CD103 expression was necessary for CD8 T cell mediated damage to the kidney in a graft model suggesting CD103 may enhance cytotoxicity.

Reviewer #2

(Remarks to the Author)

In this work, Schmidt and colleagues investigate the impact of lung inflammation on the induction of lung resident T cells. TRMs were induced in the lungs using H1N1 influenza virus or adenoviruses that express the influenza HA and NP, along with the IL-1beta cytokine. Differences in phenotype, transcriptome (sc) and persistence were observed depending on the mode of induction. TRMs induced by both modes were protective during reinfection and secondary inflammatory events in the absence of cognate antigen did not influence fate of pre-existing TRM.

The manuscript is written very well, and the conclusions are supported by the data. This work is highly relevant as it sheds light on the heterogeneity in the TRM population induced by vaccination or infection, and as such contributes to a better understanding of this cell type that is believed to provide protection during reinfections.

I have some remarks written below:

Several figures show row numbers or symbols that are displaced in the PDF file

Although the main goal of this work was to investigate the difference of Adenovirus- versus IAV-induced TRMs, the negative control (untreated animals) is not ideal. A better negative control would be mice treated with vehicle only to control for effects due to nasal instillation of vehicle.

To make the claim that inflammatory conditions are important for TRMs functionality later on, the manuscript would benefit from a characterization of the cytokine environment after infection and vaccination, so that this can be linked to the differences seen in the TRM compartment later on.

Have the authors tried to transfer TRMs from vaccinated or infected mice to naive mice and tried to correlate that with protection during secondary infection with X31?

The observation of a Tc17 cluster is very interesting and although it is only found in one mouse, do the authors think this

cluster is specific for the vaccine treatment or is there another explanation?

Did the scRNAseq analysis reveal any signatures that can explain the differences in frequencies during the memory phase seen between the vaccinated and infected groups?

CD103+ CD8+ T cells have also been described recently in the lung draining lymph nodes post influenza challenge. Did the authors consider looking at differences in the mediastinal lymph nodes of mice that were given influenza virus or adenovirus-based vaccine and how this correlates with the observations made in the lung?

Cross-reactive antibodies against NP have also shown protection in the mouse model (Lamere et al). Did the authors measure anti-NP humoral responses post infection and vaccination and can this explain the similar levels of protection during X31 reinfection?

Line 67: homologous -> homologue?

Line 668: previuos -> previous

Version 1:

Reviewer comments:

Reviewer #1

(Remarks to the Author)

The authors have generally responded positively to the critiques of their manuscript. In particular, they have addressed some of the readability issues with the previous figures. However, in many cases, they appear to have just enlarged the figures on the printed page. The original very small font sizes used appear to have been adjusted, and the figures are more readable. They also addressed the issues with nomenclature and explicit descriptions of the cell phenotypes and markers used rather than applying adjectives that are less informative. Several corrections and clarifications to the data and interpretation have been made to improve the accuracy of the report. The most substantial addition to the paper is the cytokine data that characterize the differences in the inflammatory responses. This really adds substance to the manuscript and supports the overall conclusions.

While this reviewer respects the author's choices to focus on the events in the lung rather than the lymphoid organs, the omission of analyzing the cells where they are likely being generated diminishes the strength of the paper overall and the arguments about the different immunization strategies. Quantitative immune studies in intact mouse models always benefit from a more comprehensive assessment. The intact animal is a dynamic system and the immune cell populations in different tissues are linked by migration. It is relatively little extra effort to collect and analyze tissues like draining lymph nodes and spleen when sacrificing whole animals. The cells in these organs are more plentiful and can act as controls for tissues like the lung that are more difficult to collect and process. It is much more costly in time, effort, and money to have to repeat experiments to obtain more tissues than it is to do the full analysis in the first place, especially with long-term "memory" studies. I hope the investigators take this message to heart. Without that additional data, this reviewer is left wanting but will stop short of requesting the experiments be repeated.

Reviewer #2

(Remarks to the Author)

My concerns have been addressed. The new experiments have made the manuscript even more complete.

REVIEWER COMMENTS

Reviewer #1 (Remarks to the Author):

This manuscript reports the results of studies comparing recombinant adenovirus vector vaccination (rAV) to influenza infection for the priming and development of tissue-resident memory T cells. Longevity of the T cells and protective capacity were measured. They further explore the effects on specific activation and non-specific inflammation on the prevalence and persistence of the virus-specific T cells in the lung tissue. Priming with rAV or H1N1 influenza led to distinct T cell phenotypes and gene expression profiles. Of particular note, are the observations that non-specific inflammation in the forms of LPS or heterologous influenza infection do not significantly impact the presence of TRM, while antigen specific activation alters the phenotypes and functions of the responding T cells. In general, the experiments appear to be well controlled and of good quality. The data is carefully presented. There are a few issues with the readability of the figures however.

We thank the reviewer for the overall positive evaluation of our manuscript and the constructive suggestions, which helped to further improve the quality of our study.

The authors discuss non-lymphoid tissue sites, “RAMD” in particular, and local tissue sites as a primary niches and mechanisms for generating TRM. However, there are at least two competing and non-exclusive proposed mechanisms for generation of TRM, one systemic and one tissue localized. These are probably complimentary mechanisms and should be included in the discussion. This does not minimize the significance of the tissue sites.

Thanks for this comment. We are aware of the two proposed mechanisms of “local” and “systemic divergence”. We think our data of different inflammatory milieus (see also new data set on lung cytokine expression, Suppl. Fig. 1+2) supports more the “local divergence” model, in which the fine-tuning of the TRM programming occurs within the tissue. But of course, this does not formally exclude that there were already a precursor TRM population before in the lymph node. Therefore, we could not finally confirm one or the other model, and discussed that limitation in the revised manuscript as suggested.

In many of the figures, the axis labels and much other text is too small to be easily readable. Even enlarged on screen, the figures are hard to read (for example distinguishing a “+” from a “-” superscript indicating positive and negative cells and other super- and sub-script text). There is plenty of room in the figures to enlarge the text.

Thank you for this very important comment. We refined all figures and enlarged the text for a better visualization.

In figure 1, the total numbers and proportions of HA and NP pentamer+ CD8 T cells are depicted in B and C. The reviewer appreciates the reference to the explicit surface phenotypes of the tissue subsets defined by CD69 and CD103 in C. The use of KLRG1 and

CD127 is less explicit. The reader must go to the supplemental figures and legends to find it. Can the authors more clearly define the Teff, Tem and Tcm, perhaps including in the figure legend how they used CD127 and KLRG1?

This is a very good suggestion and makes reading a bit easier. Thanks. We included a more detailed definition of the different subsets in the figure legend.

Among the Teff, Tem, Tcm, are any of those expressing CD103 or CD69?

Likewise, among the CD69 and CD103 single positives, were any of those expressing Teff, Tem, or Tcm phenotypes? Also, if I understand correctly, B presents the total number of pentamer+ cells, but the numbers below the pie charts in C also indicate total HA and NP specific cells. Those numbers are quite different, so why is that? Some of these comments may also apply to figure 5.

Thank you for the careful reading and this observation. It is true that the numbers of B and C do not exactly match due to the more stringent definitions of TRM, TEFF, TCM and TEM in C. There are of course also some cells which show phenotypes not clearly fitting in those marker combinations, e.g. iv- CD69- CD103-. Those are included in B, but not in C. However, we think this might be due to technical limitations or restriction, e.g. background staining of Pentamers on non-IAV specific cells (naïve or memory) and a further discrimination of the other population without knowing their meaning would rather confuse the reader than help.

We know that some of the iv+ cells also express for example CD103. However, our main intention was to show that lung TRM are different depending on the type of stimulation, which is clearly visible and therefore, we would like to keep the definitions and the shown population as they are. The same is true for Suppl. Fig. 5. We hope for the reviewer's /editor's understanding in this point.

The authors mention there are increase numbers of pentamer+ CD8 T cells in the rAd versus the H1N1 infected mice (line 344-347), yet there is no data shown that supports this claim.

There are statistically more HA+Pent+ at 35dpi. But the referenced passage by the reviewer was a bit misleading by mixing up statement to Pent+ and cytokine-producing cells (Fig.2). We rephrased this part for more consistence.

Does rAd result in more memory cells systemically or just more resident cells in the lung? Getting the numbers from the mediastinal lymph node and spleen would clear this up.

Since our focus lays on the characterization of TRM populations induced in the same tissue by two different means, rAd vs. IAV, we did not follow the systemic responses, which would have been also very interesting. We totally agree to the reviewers point, but unfortunately, this would mean to repeat the whole longitudinal sampling with LN or spleen samples, which is just not possible.

Furthermore, we have shown in the previous work that the systemic memory response after the intranasal immunization is less pronounced than the TRM response (Lapuente et al, 2018). Additionally, we have performed a direct comparison of vaccine-induced and rAd-induced memory T-cells from spleens at day 28 after treatment and do not see

increased numbers within the rAd-treated group (see below). However, this is only a single time point and does not allow conclusions on the long-term immunity.

The authors were initially careful to define the T cells by their surface expression patterns, rather than just applying labels. When discussing the CD4 T cell data, the authors use iv labeling, CD44, CD69, and CD103 to phenotype the cells. Beginning on line 358, the term “tissue-resident” is used interchangeably with iv-negative status. Being iv-negative does not define tissue residency, only that the cells are not in proximity to the vasculature. On line 363, the cells are referred to as “non-circulating”. There are no doubt some circulating memory cells as well as resident cells among the iv-negative population. Why not just say iv-negative, CD44+, CD4+ T cells and leave out the pejorative residency adjectives? Residency can only be demonstrated in parabiosis type experiments. Accurate terminology is important.

Thanks for this very good point. Of course, real residency can only be proven by parabiosis and therefore, we followed the suggestion and revised our manuscript in this part and only refer to iv+ or iv- status for the CD4 T-cells. We also removed the terms circulating and tissue-resident from the figures describing the ICS results. However, for the CD8 compartment, there is some evidence that the intravascular labeling is a quite good indicator for tissue-residency, we would still describe our iv- cells as TRMs. But again, we omitted the term of non-circulating as part of the TRM description, which might be not totally clear.

CD103 is the most commonly used marker of CD8 TRM, but it is not a good marker of CD4 TRM. Some have said CD4 TRM do not express CD103 at all. The data in the paper support that as most of the iv-neg (aka “tissue resident” in this paper) are CD103-. While the results of the CD4 analysis are a minor aspect of this work, it is still less ambiguous to

refer to the cells populations by their specific phenotypes rather than applying interpretive labels.

See comment above.

Can the authors better define how they determined what was a bronchiole? The images in figure 3 indicate the location of so-called bronchioles, but some look no different than the non-bronchiolar tissue. I trust their analysis did include some specific criteria, but that isn't mentioned.

Thanks for this comment. It is true that the pictures do not allow an accurate differentiation between the different structures, such as alveoli, bronchi or bronchiole. For the description of the differences between the treatment groups, it is not essential to define the Br here. Therefore, we omitted the labeling from all those pictures.

Figure 6D please label the HA, NP, and M peptides consistently with 6C labeling. It was confusing.

Thanks for the comment, but in this regard, it would be not correct to use the same labeling. In C, we only used a specific Pentamer with a defined peptide sequence and for the in vitro restimulation, we used a cocktail of peptides for the respective antigens as described in the method section.

Line 465, this may be semantics, but when discussing the numerical differences between the earlier and later longitudinal studies, "stronger" is not as precise as saying "increased numbers of". Stronger can represent both number and effector function.

We rephrased that part as suggested.

Regarding the discussion of the scRNAseq results. It was noted that a putative Tc17 population was identified as cluster 3. In line 505, a sentence starts out saying "as the only TRM population..." Is that a typo? Why is cluster 3 the only TRM population? (even if it's from only the one odd mouse)

Sorry for this inconvenience, this was a mistake. Of course cluster 3 is not the only TRM population, also not in this one mouse. The intention was to say that cluster 3 is the only TRM population not expressing typical CTL markers, such as Gzmb or Prf1. We rephrased this part for more clarity.

Figure 5: The authors classify NP pentamer-specific CD8+ T cells within their scRNA-seq dataset as either corresponding to Tcm or Trm populations, yet it is highly likely that there are T cells that are not either Tcm or Trm in the lung as well, in part based on the data represented in earlier figures.

The annotation would be more convincing if the authors would show 1) expression of selected memory subset marker genes overlayed on the shown UMAP plots and 2) heatmaps with per-cluster differential gene expression for the selected marker genes.

Indeed, if you have a closer look to Fig.1C, there are very few cells defined outside of the TCM and TRM subsets, respectively. We do not find considerable number of *Klrg1* and *Cx3cr1* expressing cells in the scRNA for example, which would be typically markers for TEM. We could clearly identify the TCM population by high *Ccr7* and *Sell* expression (cluster 5, both groups represented) and found heterogeneous expression of typical TRM markers in all other clusters, which did not allow us to further differentiate here. We took all these cells depending on the treatment group and performed DEG analyses. We agree that this might not only be TRM and added this limitation of analyses to the results.

We followed the reviewer's suggestion to show selected characteristic memory marker genes for the assigned cluster in the revised fig.7. We selected molecules out of transcription factors, effector molecules and surface or migration-related receptors. This nicely recapitulate the heterogeneity in the TRM populations of both groups. In addition, we kept the DEG analyses for the non TCM compartment as Fig. 7C together with the tables provided as supplement.

Additionally, there are several treatment-specific clusters (e.g. C0, C3, C8) which are minimally discussed elsewhere in the text. The authors should consider reanalyzing these data to perform cluster-specific annotations of populations that may differ between treatment groups – these analyses could also include specific visualizations of transcript expression for the cytokines profiled in Figs. 2 & 5.

With the new cluster-specific annotations of the selected genes, we tried to put more emphasis on the heterogeneity within the groups, but also between the two groups. The comparison of the non-TCM were mainly driven by the specific cluster C2+4 (H1N1) and C8 (rAd). However, we rephrased this part for more clarity. We also included the effector molecules, which was really helpful to visualize the difference between H1N1 and rAd in this regard and confirmed our flow data. Of course, there might be lots of different way to analyse and present the scRNA data, but we think that the new version is quite helpful. Furthermore, the data will be fully available to the scientific community for further re-analyses.

The color scheme for the heatmap depicted in figure 7E is not conducive to visualizing the weaker positive expression levels for some genes. *Itgae* is a good example. Perhaps it's just my eyes but it's a challenge to distinguish slight differences in shades of light blue. Were other color schemes tried? Red to green is often used, but obviously isn't great for those who are colorblind.

We replaced the original heatmap of Figure 7E by the new heatmaps with cluster-specific gene expression levels.

In the discussion, the authors allude to a “previously unappreciated functional role for CD103” but fail to speculate on what that might be. Would they care to speculate? For example, in 2004 Donna Farber showed that CD103 expression was necessary for CD8 T cell mediated damage to the kidney in a graft model suggesting CD103 may enhance cytotoxicity.

Thanks for this comment, we extended the discussion as suggested. The unappreciated functional role was more in terms of the longevity of the response. But it is quite interesting that CD103 expression is linked to higher cytotoxicity, but our functional flow analyses suggest rather the opposite. We added this controversial observation to the discussion of the revised manuscript.

Reviewer #2 (Remarks to the Author):

In this work, Schmidt and colleagues investigate the impact of lung inflammation on the induction of lung resident T cells. TRMs were induced in the lungs using H1N1 influenza virus or adenoviruses that express the influenza HA and NP, along with the IL-1beta cytokine. Differences in phenotype, transcriptome (sc) and persistence were observed depending on the mode of induction. TRMs induced by both modes were protective during reinfection and secondary inflammatory events in the absence of cognate antigen did not influence fate of pre-existing TRM.

The manuscript is written very well, and the conclusions are supported by the data. This work is highly relevant as it sheds light on the heterogeneity in the TRM population induced by vaccination or infection, and as such contributes to a better understanding of this cell type that is believed to provide protection during reinfections.

We thank the reviewer for the very positive evaluation of our manuscript and the highlighting the relevance of our findings. Below we address the constructive suggestions and how we think improved the quality of our manuscript.

I have some remarks written below:

Several figures show row numbers or symbols that are displaced in the PDF file

We reformatted and revised the figures throughout in the revised manuscript.

Although the main goal of this work was to investigate the difference of Adenovirus-versus IAV-induced TRMs, the negative control (untreated animals) is not ideal. A better negative control would be mice treated with vehicle only to control for effects due to nasal instillation of vehicle.

We agree with the reviewer that nasal instillation of vehicle could induce some inflammatory reactions, but it is rather unlikely that antigen-specific immunity will be induced. We and others have also shown in earlier studies that treatment with non-coding rAd-empty did not lead to antigen-specific T- cell responses, which is the focus of our study. Therefore, it would be not justified to repeat the whole longitudinal analyses with a new negative control.

To make the claim that inflammatory conditions are important for TRMs functionality later on, the manuscript would benefit from a characterization of the cytokine environment after infection and vaccination, so that this can be linked to the differences seen in the TRM compartment later on.

This is a very justified critique and we performed a new, detailed kinetic study, in which we analyzed the expression levels of 13 antiviral chemokines/cytokines in the lung homogenates of immunized or infected animals at the time points 1,3,7 and 15 post treatment via multiplex, cytometric bead assay. In addition, we analyzed the infiltrating immune cells in the BALF at that time points. Both data sets were included in the revised manuscript as supplementary figures 1+2. However, a direct causal relationship between the individual cytokines/chemokines and the different TRM populations could of course not be identified. But our data supports further the hypotheses that the local inflammatory milieu might be the main driver for the different TRM programming.

Have the authors tried to transfer TRMs from vaccinated or infected mice to naive mice and tried to correlate that with protection during secondary infection with X31?

Indeed, we tried to transfer TRM in different modalities, but without real success. The intake of the transferred cells was always very, very poor and does not allow any reliable conclusions on the functionality of our TRM.

The observation of a Tc17 cluster is very interesting and although it is only found in one mouse, do the authors think this cluster is specific for the vaccine treatment or is there another explanation?

This is highly speculative, but we do think this might be in relation to the initial expression of IL-1beta and the reduced type I interferon production. However, why this is only visible in one animal and if there is a certain threshold to be reached for this TC17 program, we cannot say.

Did the scRNAseq analysis reveal any signatures that can explain the differences in frequencies during the memory phase seen between the vaccinated and infected groups?

There are different signatures for the vaccine- or infection-induced TRM populations as described in the result part and discussed in line 464ff. However, this was just a single time point and it was not possible to correlate a specific expression profile with the long-lived TRM after vaccination. However, there might be other ways to analyze the data by the scientific community and we will be happy to share all sequencing data via an open-access archive.

CD103+ CD8+ T cells have also been described recently in the lung draining lymph nodes post influenza challenge. Did the authors consider looking at differences in the mediastinal lymph nodes of mice that were given influenza virus or adenovirus-based vaccine and how this correlates with the observations made in the lung?

Since our focus lays on the characterization of TRM populations induced in the lung by two different means, rAd vs IAV, we did not follow the responses in the LN, which would have been also very interesting. We totally agree to the reviewers point, but unfortunately, this would mean to repeat the whole longitudinal sampling with LN samples, which is just not possible.

However, we have performed a direct comparison of vaccine-induced and rAd-induced memory T-cells from spleen at day 28 after treatment and do not see increased numbers

of systemic memory cells within the rAd-treated groups (see above, answer to reviewer 1). However, this is only a single time point and does not allow conclusions on the long-term immunity.

Cross-reactive antibodies against NP have also shown protection in the mouse model (Lamere et al). Did the authors measure anti-NP humoral responses post infection and vaccination and can this explain the similar levels of protection during X31 reinfection?

Thanks for this important comment. We do show significant higher NP-specific antibodies after H1N1 infection compared to rAd immunization before the challenge (Suppl. Fig.8) and have now also added this as potential correlate of protection to the discussion. We also added the given reference of Lamere et al.

Line 67: homologous -> homologue?

Line 668: previuos -> previous

Corrected